# Role of intestinal flora in the development of nonalcoholic fatty liver disease in children

Jing Zhang,[1] Mengxuan Shi,[2] Chunna Zhao,[1] Guangcai Liang,[2,3] Chuan Li,[2,3] Xiaomeng Ge,[4] Caixia Pei,[2] Yawei Kong,[1] Dongdan Li,[1] Wenli Yang,[1] Bingyan Cao,[1] Libing Fu,[1] Yinkun Yan,[1] Jie Wu,[1] Jin Zhou,[1] Yongli Fang,[1] Xi Meng,[1] Yong Li,[5] Liming Wang[2]

**ABSTRACT** In China, 45% of adolescents with obesity develop fatty liver disease, a condition that increases the long-term risk of developing cirrhosis and liver cancer. Although the factors triggering nonalcoholic fatty liver disease (NAFLD) vary in children, the composition of intestinal microflora has been found to play an increasingly important role. However, evidence is limited on the prevalence of nonalcoholic fatty liver (NAFL) and nonalcoholic steatohepatitis (NASH) in Chinese children. Therefore, this study aimed to evaluate the fecal microbiome of Chinese children with NAFLD and further analyze the potential of flora in regulating NAFLD-related symptoms and metabolic functions. Specifically, the study applied a 16S rRNA and metagenomic sequencing to the fecal samples of pediatric patients with NAFLD, NASH, and NAFL, as well as healthy controls, to explore the correlation among NAFLD-related indexes, metabolic pathways, and gut flora. The findings showed that some fecal microbiota had a negative correlation with body mass index, and various NAFLD-related bacteria, including *Lachnoclostridium*, *Escherichia-Shigella*, and *Faecalibacterium prausnitzii*, were detected. Consequently, the study concluded that the variation in gut microbiota might be more important in improving NAFLD/NASH compared with single species, providing a microbiota diagnostic profile of NAFLD/NASH.

**IMPORTANCE** This study aims to characterize the gut microbiota in Chinese children with nonalcoholic fatty liver disease (NAFLD) through 16S rRNA and metagenomic sequencing. The results highlight the association between fecal microbiota and NAFLD in Chinese children, demonstrating distinct characteristics compared to adults and children from other countries. Based on the sequencing data from our cohort's fecal samples, we propose a microbiota model with a high area under the curve for distinguishing between NAFLD and healthy individuals. Furthermore, our follow-up study reveals that changes in the relative abundance of microbial biomarkers in this model are consistent with variations in patients' body mass index. These findings suggest the potential utility of the microbiota model and microbial biomarkers for diagnosing and treating NAFLD in children.

**KEYWORDS** BMI, children, gut microbiota, metabolism, NAFLD, NASH

O ver the past two decades, the incidence of nonalcoholic fatty liver disease (NAFLD) has more than doubled among children and adolescents globally, emerging as the most prevalent cause of chronic liver disease in various nations (1). In China, it is estimated that 45% of adolescents with obesity have NAFLD (2). According to Chinese and US clinical guidelines, children with NAFLD are more likely to develop serious diseases in adulthood, such as hypertension and liver cancer (3). NAFLD in children comprises two distinct pathological states—nonalcoholic fatty liver (NAFL) and nonalcoholic steatohepatitis (NASH)—based on the classification of liver biopsy findings

Address correspondence to Liming Wang, wanglm@im.ac.cn, Guangcai Liang, lgc_bio2021@126.com, or Jing Zhang, zhj666681@163.com.

Jing Zhang, Mengxuan Shi, and Chunna Zhao contributed equally to this article. The author order was determined in order of decreasing seniority.

The authors declare no conflict of interest.

See the funding table on p. 18.

(4). Patients with NASH face reduced survival due to cardiovascular and liver-related causes (5).

Although the factors contributing to the severity of the disease remain poorly understood, the composition of gut microbiota has emerged as a pivotal element in NAFLD pathophysiology in children. The gut microbiome comprises trillions of microbial organisms that exist symbiotically with the host, providing critical immune development, mucosal protection, and nutrient functions (1). Previous research suggests that the gut microbiome may influence the processes involved in the onset and progression of NAFLD, such as choline metabolism, obesity, endotoxemia, and liver inflammation or fibrosis (2–4). As shown in Table 1, the characterization of NAFLD/NASH gut microbiotas demonstrated varying age- and country-specific patterns.

Despite studies of the relationship between NAFLD and gut bacteria in animal models and humans (7, 10), limited research has been conducted on this issue in children, particularly concerning NAFL and NASH. Hence, this study aimed to assess the fecal microbiome of Chinese children diagnosed with NAFLD, thereby enabling the exploration of the potential of gut flora in controlling disease-associated symptoms and metabolic functions.

## MATERIALS AND METHODS

### Participants

A total of 114 participants were recruited for the study between January 2019 and December 2021, comprising 79 patients with NAFLD and 35 healthy controls (HCs). The participants were aged 7–18 years, with information on various parameters such as sex, birth date, height, and weight collected from their professionally trained parents. Recruited in the NAFLD group were patients whose body mass index (BMI) was higher than the 95th percentile and defined through a clinical evaluation and liver ultrasonography (10). The BMI values were used to classify overweight and obesity among participants, with BMI calculated as the ratio of weight (kg) to height squared ($m^2$). Participants with a recent history of antibiotic intake within 2 months or probiotics use within a month before enrollment were excluded. The liver function tests and lipid profile assessment were conducted on fasting blood samples, and the fecal samples were also collected (11). The diagnostic criteria for metabolic disorders of NAFLD were as follows (12): (i) central obesity or overweight; (ii) antidiabetic treatment or fasting glucose level ≥ 7.0 mmol/L; (iii) antihypertensive treatment or elevated collected blood pressure with ≥95th percentile in children of the same age and sex; (iv) treatment with lipid-lowering agents or triacylglycerols ≥ 1.47 mmol/L; and (v) treatment with lipid-lowering agents or high-density lipoprotein cholesterol < 1.1 mmol/L in men or <1.3 mmol/L in women. Fifteen children with NAFLD were reassessed for their BMI, and fecal samples for metagenomic sequencing were obtained following 3–6 months of nonmedical treatment such as diet control and increased exercise.

### Pathological assessment

The liver biopsies were collected from children whose parents or guardians provided consent for liver puncture to assess NAFLD activity score (NAS) and liver fibrosis stage. The liver tissues of 13 children were obtained through ultrasound-guided liver biopsy and fixed in formalin. The specimens were embedded in paraffin, after which serial sectioning was performed, and specimens were stained with hematoxylin-eosin (HE) and reticular fiber and/or Masson trichrome staining. The slides were then examined under a light microscope. Two pathologists scored the pathological diagnosis results at the same time; in case of disagreement, a senior physician was invited. NAFL was assessed according to NAS diagnosis criteria of NAFLD on pathological liver tissue (or liver biopsy) (13). NAFL was defined as liver cell steatosis without any indication of steatohepatitis, with or without liver fibrosis; NASH was defined as the presence of hepatic steatosis

**TABLE 1** Studies of microbiome differentiation in patients with NAFLD

| Studies | Groups | Methods | Microbiome |
|---|---|---|---|
| Lee et al., 2020 (1) | 171 patients with NAFLD and 31 non-NAFLD controls (adult Korean) | Fecal microbiome: 16S rRNA | NAFLD vs healthy control (HC): *Faecalibacterium*↓ Ruminococcaceae↓ Lachnospiraceae↓ *Coprococcus*↓ Rikenellaceae↑ Enterobacteriaceae↑ *Citrobacter*↑ |
| Adams et al., 2020 (2) | 67 patients with NAFLD and 55 non-NAFLD controls (adult Australian) | Fecal microbiome: 16S rRNA | NAFLD vs HC: Lachnospiraceae↑ Bacteroidaceae↓ |
| Boursier et al., 2016 (3) | 57 patients with NAFLD, including 35 with NASH (adult French) | Fecal microbiome: 16S rRNA | NASH vs NAFLD: *Bacteroides*↑ *Ruminococcus*↑ *Prevotella*↓ |
| Marialena et al., 2013 (4) | 11 patients with NAFLD, 22 with NASH, and 17 non-NAFLD controls (adult Canadian) | Fecal microbiome: quantitative real-time polymerase chain reaction | NASH vs NAFLD/HC: Bacteroidetes↓ *Clostridium coccoides*↑ |
| Fei et al., 2020 (5) | 32 patients with NAFLD and 36 non-NAFLD controls (pediatric Chinese) | Fecal microbiome: metagenome | NAFLD vs HC: Ruminococcaceae↓ *Faecalibacterium*↓ |
| Monga et al., 2020 (6) | 73 patients with NAFLD (pediatric American) | Fecal microbiome: 16S rRNA | HFF>5.5% vs HFF<5.5%: *Prevotella*↓ *Gemmiger*↓ *Oscillospira*↓ |
| Schwimmer et al., 2019 (7) | 87 patients with biopsy-proven NAFLD and 37 obese non-NAFLD patients (pediatric American) | Fecal microbiome: 16S rRNA | NASH vs NAFLD: *Faecalibacterium*↓ *Alistipes*↓ *Parabacteroides*↓ |
| Federica et al., 2017 (8) | 27 patients with NAFL, 26 with NASH, or 8 obese and 54 healthy controls (pediatric Italian) | Fecal microbiome: 16S rRNA | NASH vs HC: *Oscillospira*↓ *Dorea*↑ *Ruminococcus*↑ *Blautia*↑ |
| Zhu et al., 2013 (9) | 25 obese, 22 with NASH, and 16 healthy controls (pediatric American) | Fecal microbiome: 16S rRNA | NASH vs HC: Proteobacteria↑ *Escherichia*↑ *Prevotella*↑ Lachnospiraceae↓ Ruminococcaceae↓ |

with necroinflammation and hepatocellular injury with or without fibrosis. NAFLD with fibrosis can refer to either NAFL or NASH associated with fibrosis in port, periportal, perisinusoidal, or portal-portal bridging fibrosis (13). The diagnostic criteria for NAFLD and NASH, including the scoring rules for NAS, the stage of liver fibrosis, and the pathological diagnosis of NASH, refer to the Expert Consensus on the Diagnosis and Treatment of NAFLD in Children and the study by Kleiner et al. (13, 14).

Out of the 79 children diagnosed with NAFLD, our analysis included only 13 children for whom parental consent was obtained for liver puncture. Following the evaluation of

the NAS, we identified eight cases of NASH and five cases of NAFL. As shown in Fig. 1, the NAS was created as an unweighted score for steatosis (0–3), lobular inflammation (0–3), and ballooning (0–2). A four-scale grading (from 0 to 3) was used for grading steatosis: normal liver (grade 0) contains fat in <5% of hepatocytes, while grade 1 steatosis refers to <33% steatotic hepatocytes. In grade 2 and 3 steatoses, fat is present in at least 33% or 66% of hepatocytes, respectively. The grading system of lobular inflammation was determined by the number of inflammatory foci in a given area, from grade 0 to 3 (0–3 points, none, <2, 2–4, and >4, respectively). The grading of ballooned cells was based mainly on number (0–2 points, none, few, and many, respectively). The total NAS score ranged from 0 to 8, where scores of 0–2 can exclude NASH, 3–4 suggested possible NASH, and scores of 5–8 were used to diagnose NASH. Additionally, the progression of fibrosis was described as follows (Fig. 1): zone 3 perivenular and perisinusoidal represented stage 1 (with a subdivision into 1a and 1b according to the amount of the deposit), and stage 1c was isolated periportal fibrosis; stage 2 included portal and central fibrosis without bridging fibrosis; stage 3 was bridging fibrosis; and stage 4 was cirrhosis.

## Biochemical data

After overnight fasting, 4 mL of peripheral blood was drawn from the cubital vein of each participant (15). The blood samples were centrifuged at 2,500 $g$ for 10 min using an analyzer (KHB, Shanghai, China). The resulting supernatants were collected and stored for analysis. The serum levels of fasting blood glucose, uric acid, thyroid-stimulating hormone (TSH), alanine aminotransferase (ALT), and aspartate aminotransferase (AST) were measured using a Beckman Coulter AU5800 analyzer (CA, USA).

## Fecal sample collection and DNA extraction

The fecal specimens from participants were collected in sterilized 2 mL tubes containing a protective solution and frozen at −80°C until DNA extraction. Total DNA was extracted from the fecal samples (250 mg, wet weight) using a PSP Spin StooL DNA Kit/PSP Spin Stool DNA Plus Kit (Stratec Molecular GmbH, Berlin, Germany) following the manufacturer's protocols (10).

## 16S rRNA sequencing and analysis

The stool samples were processed at MyGenostics Co. Ltd. (Beijing, China). The 16S rRNA amplicon sequencing of the V3–V4 gene region was used on an Illumina MiSeq platform to detect and quantify bacterial taxa in the fecal samples of HCs and children with NAFLD. The paired-end 16S sequences were merged and filtered for quality using

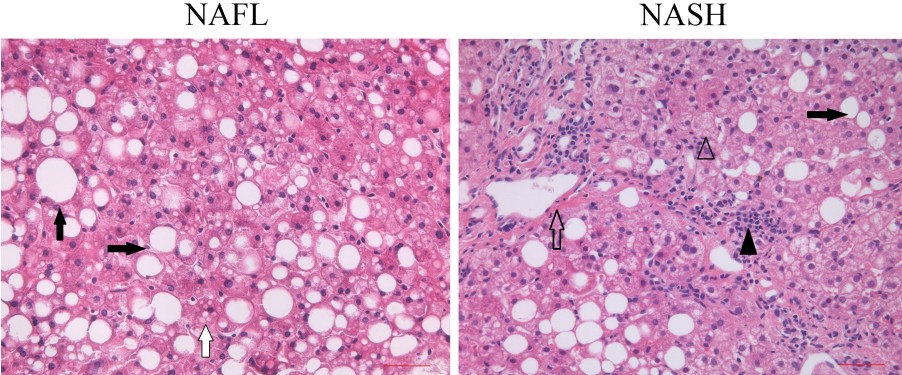

**FIG 1** The histomorphological diagnosis of liver biopsy specimens in patients diagnosed with NAFL or NASH. The HE staining of the liver biopsy specimens of NAFL only shows steatosis. The HE staining of NASH shows steatosis, lobular inflammation, ballooning, and liver fibrosis. The black arrow, white arrow, black arrowhead, empty arrowhead, and empty arrow indicate macrovesicular steatosis, microvesicular steatosis, lobular inflammation, ballooning, and liver fibrosis, respectively. Scale bar, 250 µm. Images of all pathological sections were provided by the Pathological Department, Beijing Children's Hospital.

the laser flash method. The Quantitative Insights into Microbial Ecology (QIIME, version 2.0) software suite was used to analyze the sequences, and the QIIME tutorial (http://qiime.org/) was followed with some modifications. The sequences were compared with the 97% reference data set of the SILVA database (SILVA project, version 138.1, http://www.arb-silva.de/). Those that did not match any reference sequences were clustered into *de novo* operational taxonomic units (OTUs) at 97% similarity using the UCLUST algorithm. This allowed us to accurately assign taxonomical classifications to our sequencing reads and perform downstream analyses with confidence. To obtain the population profiles of common and abundant taxa, we constructed a distance matrix based on the abundance of OTUs. Hierarchical clustering was performed using the unweighted pair group method with arithmetic mean (average linkage clustering) on this distance matrix. A Newick tree format was obtained using the QIIME package. Furthermore, QIIME was used to analyze the alpha diversity (observed species, Chao1, Shannon, and Simpson) and beta diversity [principal coordinate analysis (PCoA)]. Linear discriminant analysis effect size (LEfSe) was applied to compare significant differences in taxa between patients with NAFLD and HCs using a $P$ value < 0.05.

In this study, we utilized the random forest supervised learning algorithm on 16S data sets to distinguish between HCs and NAFLD samples. We set the ntree parameter to 500 for the number of trees to construct. Two-thirds of all data were utilized as the training set, while the remaining third constituted the test set. To reduce the possibility of confounding variables due to correlated data, we normalized the filtered relative abundances. The model was trained using 114 stool samples, 35 of which were HC and 79 were NAFLD samples. The class error calculated using the model was 0.3714. The model accurately identified 22 HC samples. For the NAFLD samples, the model could predict all samples precisely. The estimated error rate of the model was 0.11404, while the baseline error for random guessing sat at 0.30702. The ratio of baseline error to observed error for our modeling was 2.69231. We used the R programming language to calculate both area under curve (AUC) and the top 10 key indicators generated by the model.

## Metagenomic sequencing and analysis

To investigate the bacterial species spectrum associated with NAFLD clinical diagnosis, we conducted metagenomic sequencing on 79 fecal samples obtained from children with NAFLD. The stool specimens were submitted to MyGenostics Co. Ltd. (Beijing, China) for analysis. During metagenomic sequencing, 20 µg of microbial metagenomic DNA was extracted and purified. Also, a sequencing library was prepared using a NEXTflex Rapid DNA Sequencing Kit (NOVA-5144-02, Bioo Scientific, TX, USA) following the manufacturer's protocols. After library preparation, Qubit 2.0 was used to preliminarily quantify the library, which was then diluted to 2 ng/µL. Agilent 2100 was used to detect the insert size of the library, and the quantitative polymerase chain reaction method was used to accurately measure the effective concentration of the library (>3 mmol/L) to ensure the quality of the library. The constructed library was sequenced on an Illumina NovaSeq 6000 platform. MOCAT preprocessing was employed to ensure the quality of data, which involved filtering out sequences with low quality, insufficient length, or containing errors. The resulting clean data from each sample were then used for metagenomic assembly. Subsequently, all the samples were assembled, with the objective of generating reads longer than 300 bp for further analysis. The genes were combined and clustered using CD-Hit software, ultimately resulting in 2,053,172 nonredundant genes.

The gene catalog was used to compare the sequence with the MicroNR library to obtain species annotation information of unigenes employing bowtie. Subsequently, the gene abundance table was merged to generate a species abundance table with different classification levels. The overall distinctions in microbial composition were evaluated by PCoA analysis of the Bray-Curtis distance. LEfSe was performed, and a significance level of $P < 0.05$ was used to detect significant variations in taxa between patients with NAFL and NASH. The sequences obtained from metagenomic sequencing and

the Kyoto Encyclopedia of Genes and Genomes (KEGG) database were compared using BLAST (version 2.2.28+) to identify the metabolic functions of gut microbiota within the samples. The predicted KEGG Orthology-Based Annotation System 2.0 was used for this purpose. The relationship between the structure of intestinal microbiota and NAFLD-related indexes was analyzed using Spearman's correlation.

## Statistical analysis

The present study used SPSS 22.0 software and an R environment during statistical analysis. Before conducting the statistical analyses, the normality was evaluated for all numerical variables per previous recommendations (16). Generalized linear models were employed to compare continuous data across groups. The chi-square analyses were conducted to compare frequencies among groups, and the independent-sample *t*-tests were used to compare age and BMI across groups. The Kruskal-Wallis test was used to investigate the differences in taxonomic composition and function annotations between groups. Alpha diversity was calculated using the Shannon index. In contrast, differential diversity analyses were conducted using either the Wilcoxon rank-sum test or the Kruskal-Wallis test, depending on the number of groups. Kruskal-Wallis *H*-tests and Welch's *t*-tests were used to calculate *P* values, and false discovery rates were employed to control for multiple hypothesis testing. Mann-Whitney *U* tests were used to compare the abundance of each phylum and genus among groups. For phyla and genera in which Student's *t*-tests were employed, the abundance values were ranked and used in regression models. Additionally, each multiple linear regression model was checked for normality using predicted probability plots, and the absence of multicollinearity was assessed using variance inflation factor values.

## RESULTS

### Ecological diversities and significant differences in bacterial abundance between patients with NAFLD and HCs

This study comprised 114 children between 7 and 18 years old, among which 79 were with NAFLD and 35 were HCs. Height, weight, and BMI were measured, and no significant differences were observed in sex or age between the two groups. However, the BMI of patients with NAFLD was significantly higher than that of HCs. The anthropometric and demographic data are presented in Table 2.

The gut microbiomes of 79 patients with NAFLD and 35 HCs were analyzed using 16S rRNA sequencing to explore the differences in intestinal flora diversity and the abundance of various taxa. A total of 16,570,490 sequencing reads were obtained from the 114 samples. The fecal microbiotas observed in the samples were primarily composed of the phyla Firmicutes, Bacteroidetes, and Proteobacteria (Fig. 2A). Only the abundance of Proteobacteria was found to increase in pediatric patients with NAFLD compared with HCs. The dominant genera observed in both groups were *Bacteroides*, *Faecalibacterium*, *Prevotella* 9, and *Bifidobacterium* (Fig. 2B). However, the abundance of *Bacteroides* and *Prevotella* 9 both increased in pediatric patients with NAFLD, whereas the abundance of *Faecalibacterium* decreased. Differences in the abundance of several bacteria were observed between NAFLD and HC groups, and these bacteria were planned for further analysis. No significant differences in species richness and evenness were observed

**TABLE 2** Anthropometric and demographic data in NAFLD and HC groups

| | HC (*n* = 35) | NAFLD (*n* = 79) | *P* |
|---|---|---|---|
| Sex (male/female) | 24/11 | 65/14 | 0.064 |
| Age (year) | 11.42 ± 1.08 | 12.23 ± 2.63 | 0.110 |
| Height (cm) | 151.28 ± 14.87 | 158.22 ± 14.37 | 0.234 |
| Weight (kg) | 40.14 ± 12.11 | 72.38 ± 20.01 | 0.090 |
| BMI (kg/m$^2$) | 17.17 ± 2.34 | 28.37 ± 4.36 | 0.001 |

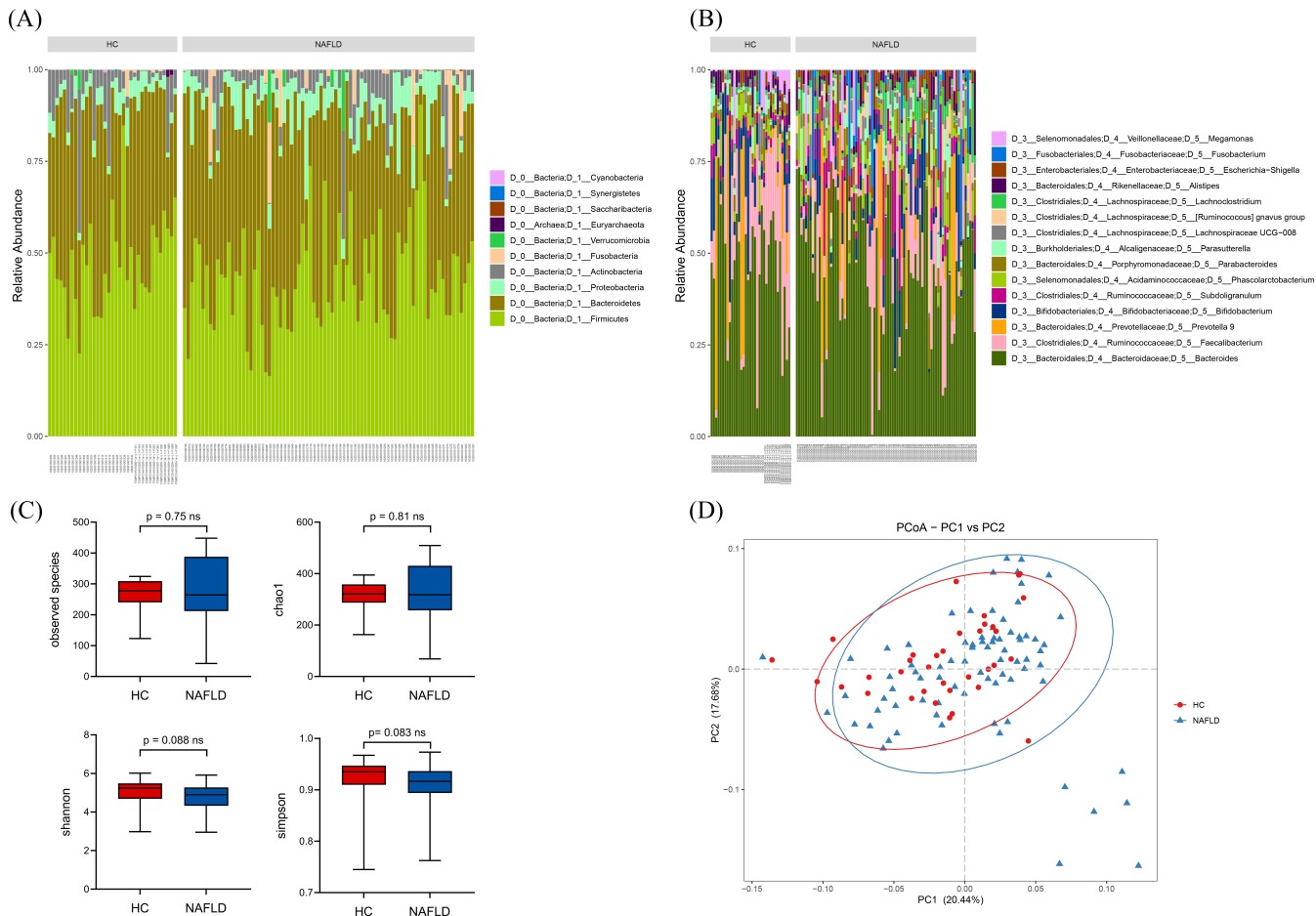

**FIG 2** Colony composition and diversity of fecal microbiota in pediatric patients with NAFLD and HCs. (A) Top 10 species at the phylum level. (B) Top 15 species at the genus level in each sample in pediatric patients with NAFLD and HCs. (C) Alpha-diversity group comparison of patients with NAFLD and HCs. (D) PCoA of patients with NAFLD and HCs (HC, *n* = 35 and NAFLD, *n* = 79).

between HCs and patients with NAFLD in median alpha diversity (observed species, Chao1, Shannon, and Simpson diversity indices) (Fig. 2C). However, the observed species in patients with NAFLD were more dispersed than those observed in HCs. A PCoA analysis of the gut microbiota samples of the children is presented in Fig. 2D. The statistical models revealed that the composition and abundance of flora species in patients with NAFLD and HCs were similar.

The analysis of bacterial abundance among groups revealed a significant difference in the abundance of several species of bacteria (Fig. 3). Specifically, the abundance of 24 species of bacteria, including Lachnospiraceae, *Proteobacteria, Ruminococcus gnavus* group, *Fusobacteria*, Fusobacteriales, *Lachnoclostridium*, Enterobacteriaceae, Enterobacteriales, and *Escherichia-Shigella,* increased in the gut microbiota of pediatric patients with NAFLD. Conversely, the abundance of 51 species of bacteria, including Ruminococcaceae, *Faecalibacterium*, Bifidobacteriaceae, Bifidobacteriales, *Prevotella* 9, Prevotellaceae, *Ruminococcus* 1, and Veillonellaceae, was significantly lower in patients with NAFLD compared with HCs (Fig. 3). Additionally, *Methanobacteria* belonging to Euryarchaeota was a characteristic genus in HCs. Conversely, *Fusobacteria* was the characteristic genus in patients with NAFLD.

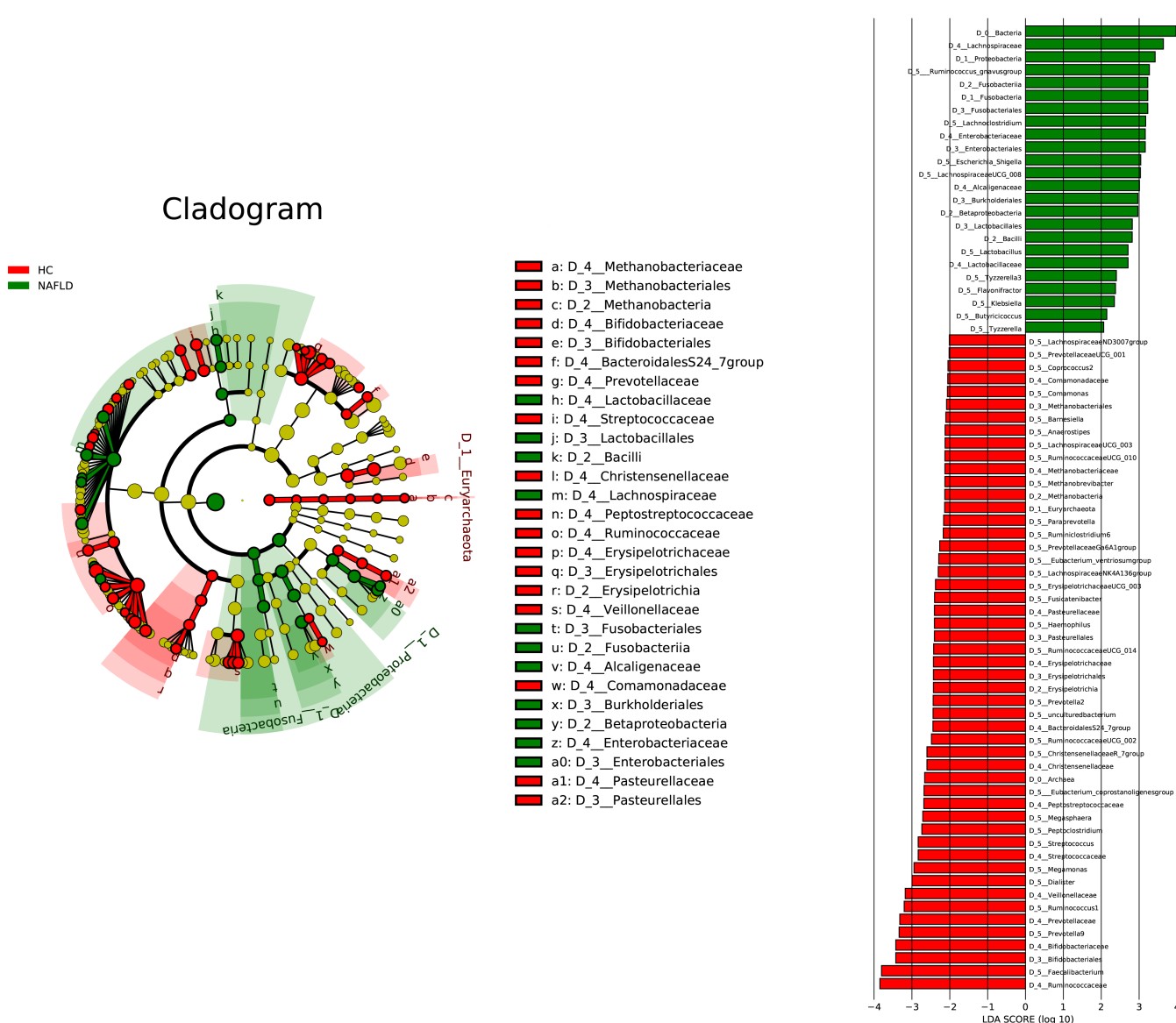

**FIG 3** Multiple bacterial species with significantly different abundance were found in the intestinal flora of patients with NAFLD and HCs (HC, $n = 35$ and NAFLD, $n = 79$, $P < 0.05$).

## Ecological diversities and significant differences in bacterial abundance between patients with NASH and those with NAFL

Eight patients with NASH and five with NAFL were identified among children who consented to liver puncture, and the pathological findings were consistent with the diagnosis of NAFLD. Patient BMI, fasting blood glucose, uric acid, TSH, and two liver enzyme indices were measured and compared between the NAFL and NASH groups. Table 3 illustrates the anthropometric and clinical features of the patients. Patients with NASH had higher levels of AST and TSH than those with NAFL.

The samples from patients with NAFL and NASH underwent metagenomic sequencing to determine the diversity and significant differences in intestinal flora between the two groups. We discovered that the number and evenness of flora species, as well as the Shannon and Simpson diversity indices, decreased in patients with NAFL compared with NASH (Fig. 4A). The non-metric multidimensional scaling (NMDS) plot of the gut

**TABLE 3** Anthropometric and demographic data in NAFL and NASH groups[a]

|  | NAFL (n = 5) | NASH (n = 8)[b] | P value |
|---|---|---|---|
| Sex (male/female) | 4/1 | 8/0 | 0.007 |
| Age (year) | 11.57 ± 2.23 | 11.6 ± 3.91 | 0.023 |
| Height (cm) | 152.43 ± 16.48 | 149.55 ± 11.89 | 0.386 |
| Weight (kg) | 69.20 ± 26.73 | 57.51 ± 14.37 | 0.464 |
| BMI (kg/m$^2$) | 28.80 ± 5.90 | 25.26 ± 2.50 | 0.153 |
| Fasting blood glucose (mg/dL) | 5.53 ± 1.30 | 4.98 ± 0.37 | 0.041 |
| Uric acid (mg/dL) | 447.01 ± 79.38 | 455.5 ± 110.4 | 0.304 |
| AST (U/L) | 54.13 ± 27.07 | 76.74 ± 61.73 | 0.591 |
| ALT (U/L) | 100.3 ± 61.12 | 94.67 ± 94.86 | 0.493 |
| TSH (uIU/mL) | 2.76 ± 1.42 | 3.68 ± 2.43 | 0.430 |
| FT3 | 6.44 ± 1.24 | 6.59 ± 0.71 | 0.305 |
| FT4 | 11.63 ± 2.13 | 11.78 ± 1.64 | 0.279 |
| NAS | 3.14 ± 0.38 | 4.10 ± 1.10 | 0.051 |
| Fibrosis stage |  |  |  |
| 0 | 2/5 | 0/7[b] |  |
| 1a | 2/5 | 0/7 |  |
| 1b | 0/5 | 0/7 |  |
| 1c | 1/5 | 1/7 |  |
| 2 | 0/5 | 4/7 |  |
| 3 | 0/5 | 2/7 |  |

[a]FT3, free triiodothyronine; FT4, free thyroxine.
[b]7 out of 8 NASH patients carried out examined of liver fibrosis.

microbiota samples revealed that the abundance and composition of flora species in patients with NASH were similar to those in patients with NAFL (Fig. 4B). Further analysis using LEfSe identified several significantly different genera and species between the two groups. Patients with NASH had a higher abundance of *Alistipes,* whereas Peptostreptococcaceae noname had a lower abundance at the genus level (P < 0.05, Fig. 4C). The abundance of *Bacteroides uniformis,* Lachnospiraceae bacterium 7_1_58FAA, *Eubacterium ventriosum*, and unclassified *Roseburia* increased significantly in patients with NASH compared with those with NAFL at the species level (P < 0.05, Fig. 4D).

## Correlation of bacterial abundance with NAFLD-related indexes

We conducted metagenomic sequencing on all 79 NAFLD samples to analyze the correlation between NAFLD-related indices and microbiota in children. The abundance of certain species positively correlated with NAFLD-related indices. The abundance of *Streptococcus mitis, Streptococcus oralis, Streptococcus pneumoniae*, *Gemella* unclassified, *Collinsella* unclassified, *Granulicatella adiacens*, and *Streptococcus cristatus* positively correlated with BMI (Fig. 5). GLU positively correlated with the abundance of *Weissella confusa,* Lachnospiraceae bacterium 2_1_58FAA, *Roseburia intestinalis,* Lachnospiraceae bacterium 6_1_63FAA, and *Paraprevotella xylaniphila* (Fig. 5). ALT positively correlated with the abundance of *Eubacterium ruminantium* group, Ruminococcaceae NK4A214 group, Erysipelotrichaceae UCB-003, *Ruminococcus inulinivorans*, and *Coprococcus comes*. AST positively correlated with the abundance of *Barnesiella intestinihominis*, *Eubacterium redale*, *R. inulinivorans*, and *Akkermansia muciniphila* (Fig. 5). Only the abundance of unclassified *Neisseria* positively correlated with TSH (Fig. 5). Uric acid (UA) negatively correlated with the abundance of *Ruminiclostridium* 9, *Collinsella aerofaciens*, and *Eubacterium eligens*, whereas *Collinsella intestinalis*, *Collinsella* unclassified, *Bacteroidetes xylanisolvens*, and *Megamonas rupellensis* showed a positive correlation (Fig. 5). These findings suggested a correlation between gut microbiota dysbiosis and NAFLD development, with these bacteria being considered NAFLD-related taxa.

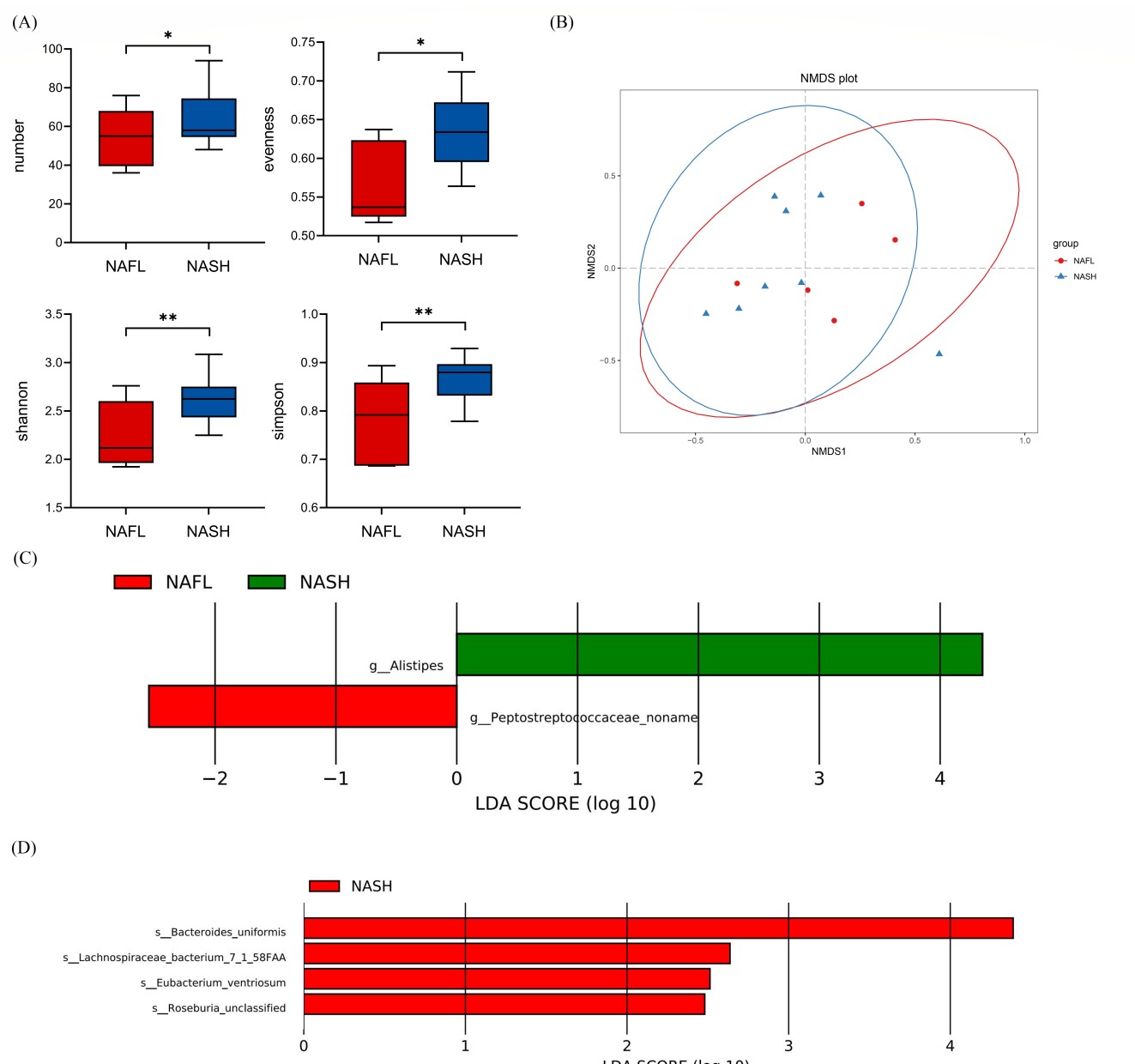

**FIG 4** Differential diversity and multiple bacterial species with significantly different abundance were found in the intestinal flora of patients with NAFL and NASH. (A) Alpha-diversity group comparison of patients with NAFL and NASH. (B) NMDS of patients with NAFL and NASHs. (C) Genera with significant differences in abundance between patients with NAFL and those with NASH. (D) Species with significant differences in abundance between patients with NAFL and NASH (NAFL, $n = 5$ and NASH, $n = 8$, $P < 0.05$).

## Correlation of bacterial abundance with NAFLD-related metabolic pathways

We conducted a further examination of disruptions in gut microflora associated with specific metabolic pathways in NAFLD. These pathways included glucose metabolism, fatty acid metabolism, short-chain fatty acid (SCFA) metabolism, and folate pathways. Utilizing metagenomic sequencing data from 79 NAFLD patients, we assessed the correlation between bacterial taxa and these metabolic pathways. As shown in Fig. 6, the abundance of *Turicibacter sanguinis, Eubacterium dolichum*, and *Bacteroides clarus* was positively correlated with xylose degradation IV (PWY-7294). The abundance of *Weissella confusa, Actinomyces odontolyticus, S. mitis, S. oralis, S. pneumoniae*, and *Streptococcus*

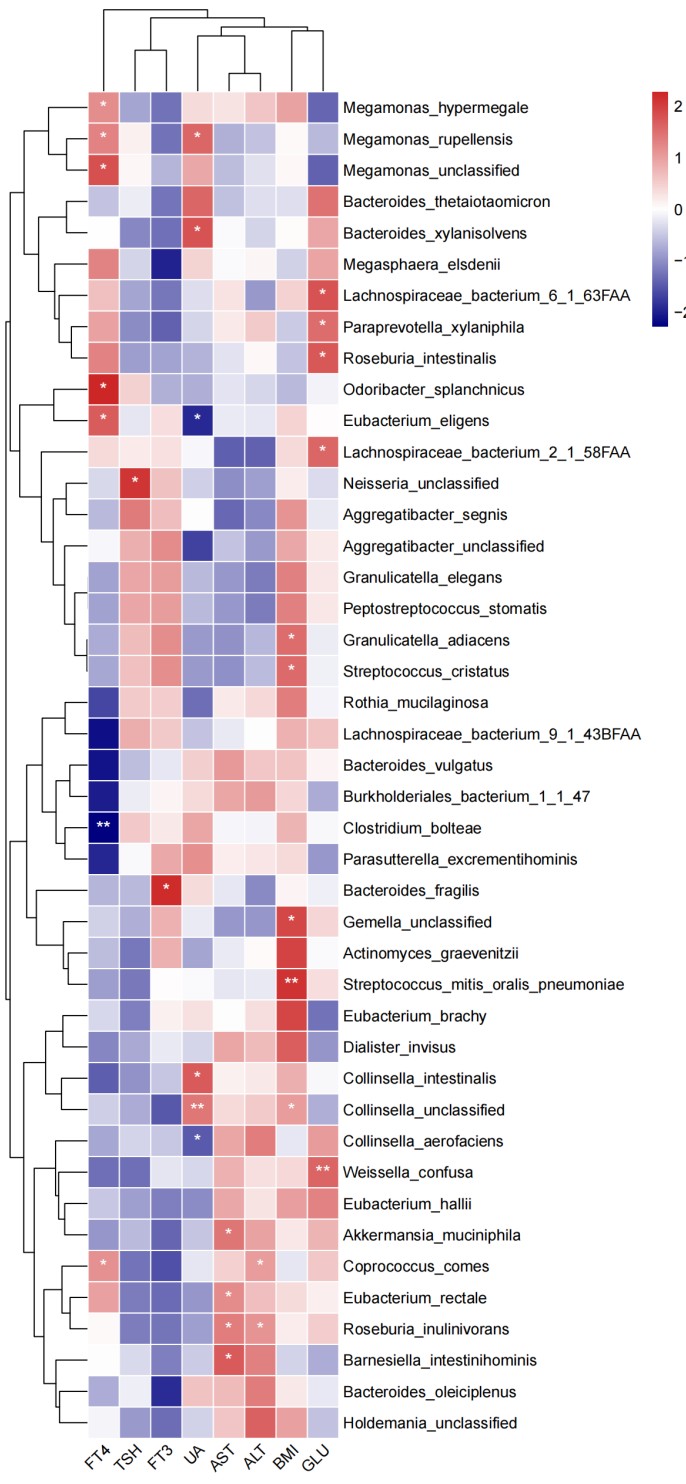

**FIG 5** Abundance of species significantly correlated with NAFLD-related indicators at the species level (*n* = 79, *P* < 0.05 and **P** < 0.01).

*australis* was positively correlated with N10-formyl-tetrahydrofolate biosynthesis and folate transformations II (PWY-3841), whereas negatively correlated with the abundance of Clostridiales noname. The abundance of *Citrobacter freundii* and *Citrobacter* unclassified was positively correlated with fatty acid beta-oxidation V (PWY-6837), fatty acid salvage (PWY-7094), and L-glutamate degradation IV (PWY-4321). The abundance of *Neisseria subflava, Neisseria flavescens, Fusobacterium periodonticum, Lautropia mirabilis,*

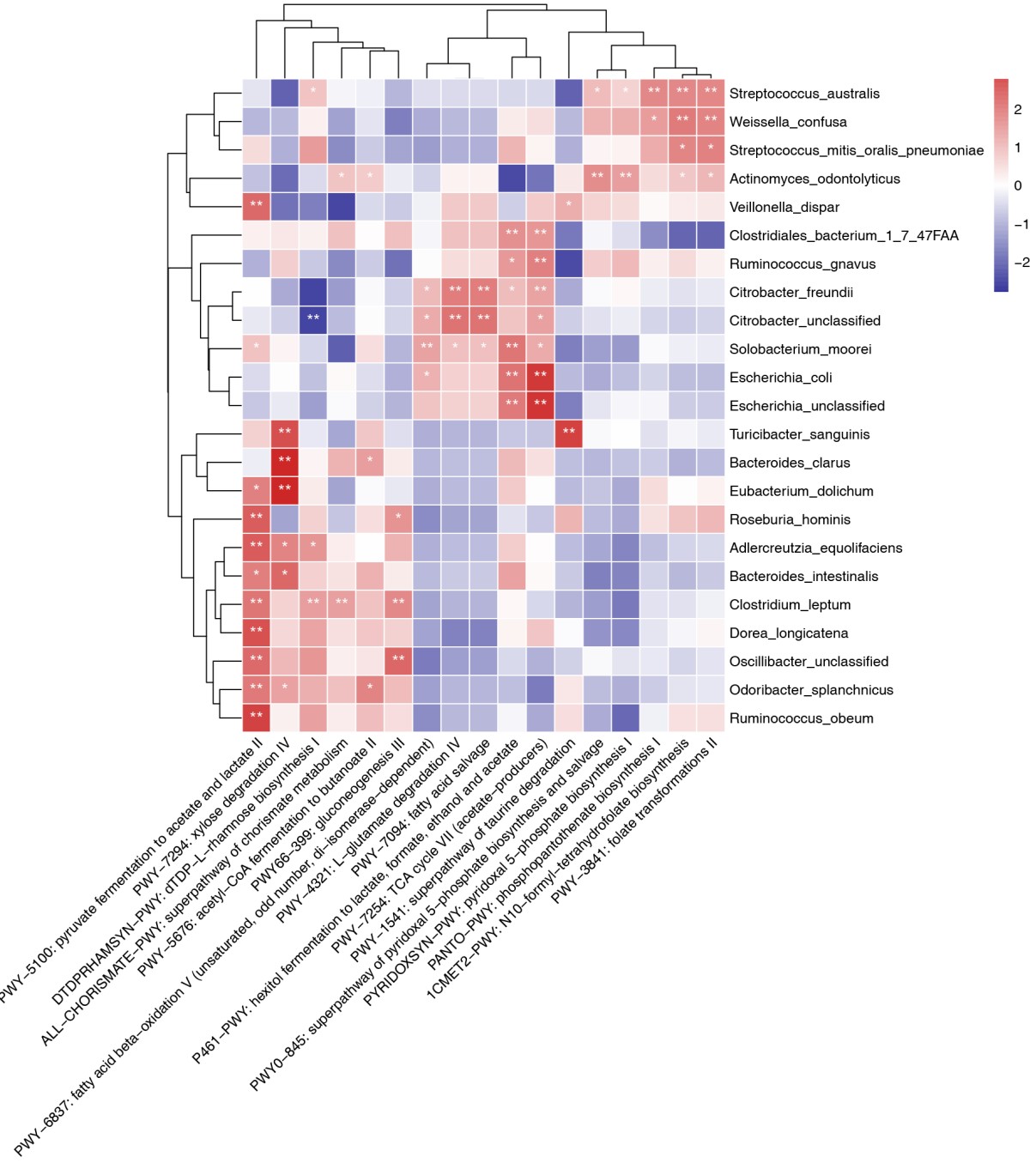

**FIG 6** Abundance of species significantly correlated with microbial metabolic pathways potentially related to NAFLD at the species level ($n = 79$, *$P < 0.05$ and **$P < 0.01$).

and *T. sanguinis* showed a positive correlation with the superpathway of taurine degradation (PWY-1541).

The abundance of different bacterial taxa was strongly associated with the metabolism of SCFAs (Fig. 6). Bacteria such as Clostridiales bacterium 1_7_47FAA, *Escherichia coli*, unclassified *Escherichia*, *Veillonella parvula*, *R. gnavus*, and *C. freundii* were found to be strongly linked to the tricarboxylic acid (TCA) cycle VII, which was involved in the production of acetate. Bacteria such as unclassified *Oscillibacter*, *Dorea longicatena*, *Adlercreutzia equolifaciens*, *Holdemania filiformis*, *Atopobium*, *Clostridium leptum*, *Roseburia hominis*, *R. obeum*, *E. dolichum*, *Odoribacter splanchnicus*, *Bacteroides intestinalis*,

and *Veillonella dispar* were strongly linked to pyruvate fermentation to acetate and lactate II (PWY-5100). The abundance of *Enterobacter cloacae* and *B. clarus* was found to be strongly associated with acetyl-CoA fermentation to butyrate. Similarly, the abundance of *Solobacterium moorei*, unclassified *Granulicatella*, and Clostridiales bacterium 1_7_47FAA was strongly linked to the hexitol fermentation to lactate, formate, ethanol, and acetate (P461-PWY). These findings indicate a close relationship between the composition of fecal bacteria and specific metabolic pathways within the gut microbiota. Perturbations in bacterial composition could potentially impact the development of NAFLD through microbiota metabolism.

## Prediction model of the identified bacterial taxa for the clinical discrimination of patients with NAFLD and HCs

We developed a stochastic forest machine algorithm to construct a prediction model that could discriminate between patients with NAFLD and HCs based on the identified bacterial taxa. We performed 16S rRNA sequencing analysis of the gut microbiomes of patients with NAFLD and HCs. Ten bacterial taxa, including *Sneathia*, Lachnospiraceae ND3007 group, *Prevotella, Lachnoclostridium, Escherichia-Shigella,* Prevotellaceae UCG-001, *Haemophilus, Lactobacillus, Ruminiclostridium* 6, and *Coprococcus* 2, were selected to predict the risk of NAFLD in children (Fig. 7A). The prediction accuracy of the model was assessed using receiver operating characteristic (ROC) curves, which are presented in Fig. 7B. The model achieved an AUC value of 0.969, with a sensitivity of 0.914 and a specificity of 0.949.

## Relationship of the NAFLD-related changes in bacterial abundance with the BMI under the regulation of nonmedical treatment

In this study, we examined the alterations in the diversity of the gut microbiota and abundance of NAFLD-related bacteria before and after nonpharmacological treatment to establish the correlation between gut microbiota and BMI reduction. Fifteen children diagnosed with NAFLD agreed to participate in the study. Five children recorded a decrease in BMI greater than 5% after the nonpharmacological intervention, and their

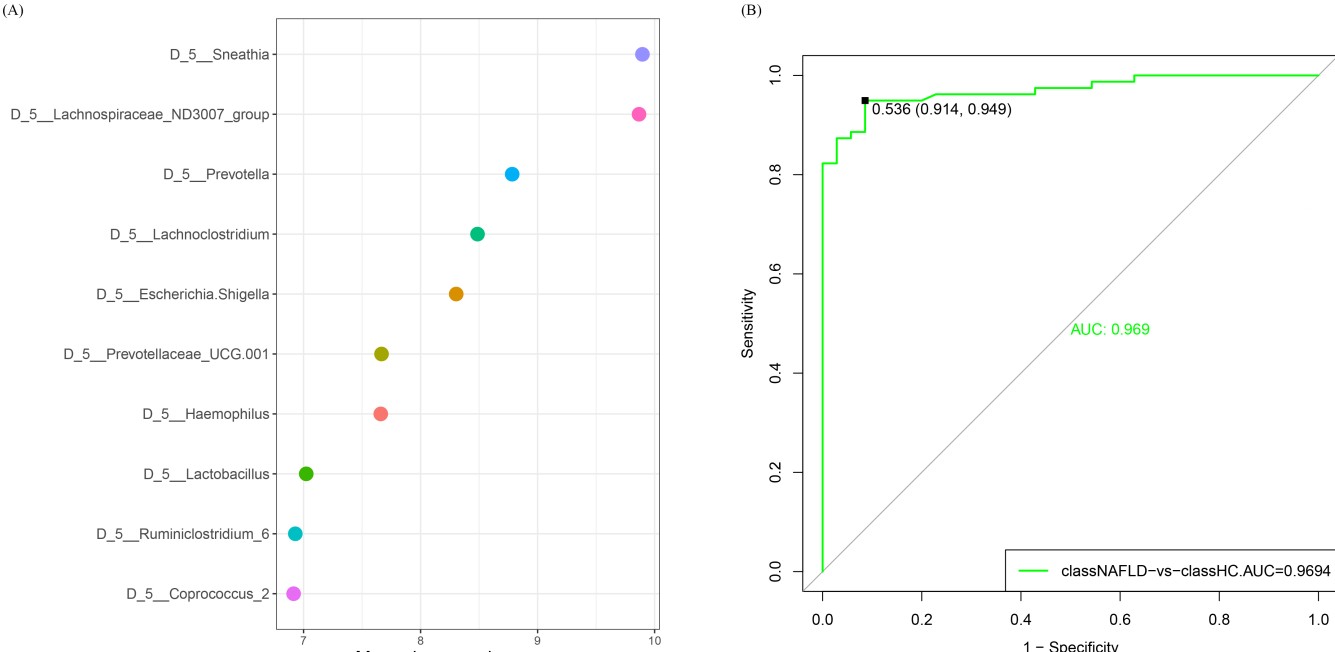

**FIG 7** A prediction model of the identified bacterial taxa for the clinical discrimination of patients with NAFLD and HCs was built using the stochastic forest machine algorithm. (A) Prediction model constructed using the stochastic forest machine algorithm and (B) ROC curves for clinical discrimination.

treatment was labeled effective. Conversely, two children had a BMI increase greater than 5%, indicating ineffective treatment. The remaining eight children showed a decrease in BMI of less than 5%, which was deemed irrelevant in this study. Subsequently, we conducted metagenomic sequencing and the resultant analysis of microbial taxa on stool samples of children whose BMI increased or decreased by more than 5%. Figure 8A displays the study results, revealing an increase in the abundance of gut flora species in children with reduced BMI after nonpharmacological treatment. In contrast, those with increased BMI experienced a decrease. The abundance of *Bifidobacterium longum* and *E. ventriosum* increased significantly with nonpharmacological treatment (Fig. 8B). Furthermore, the abundance of Alcaligenaceae, Erysipelotrichaceae, *Faecalibacterium prausnitzii*, *Subdoligranulum* unclassified, *Lachnoclostridium*, *R. intestinalis*, Lachnospiraceae UCG-008, and *C. comes* all demonstrated significant increases following nonpharmacological treatment (Fig. 8B). Conversely, the study noted a decrease in the abundance of *Haemophilus*, *E. coli*, *Escherichia* unclassified, *Ruminococcus torques*, and Lachnospiraceae bacterium 2_1_58FA in patients with NAFLD with reduced BMI after treatment (Fig. 8C). These results suggested a correlation between the diversity of gut

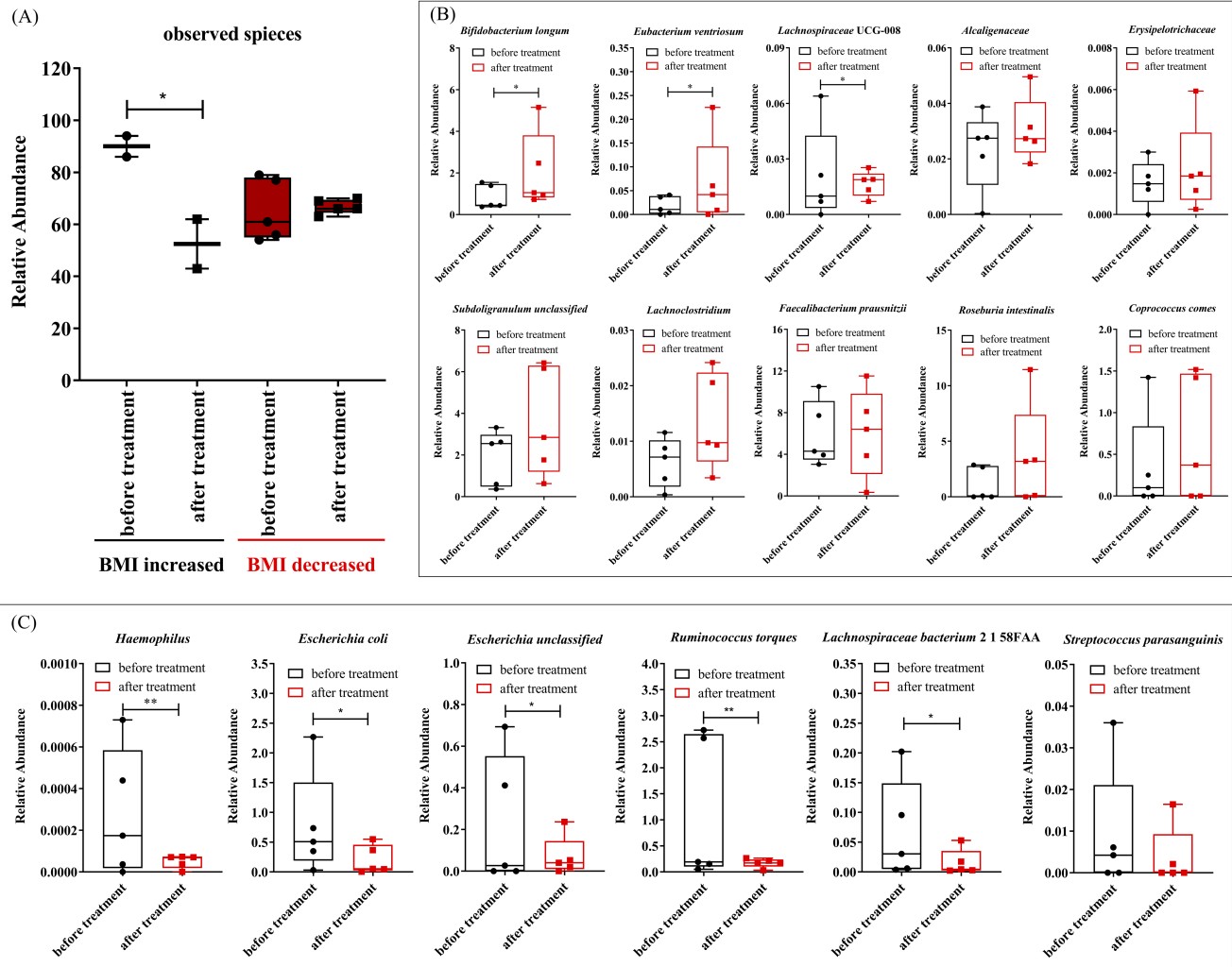

**FIG 8**  Change in gut microbiota diversity and abundance of NAFLD-related bacteria was closely related to BMI decrease in pediatric patients with NAFLD after nonpharmacological treatment. (A) Observed species of bacteria in the flora of pediatric patients with NAFLD. (B) Enhanced bacterial abundance and (C) reduced bacterial abundance after nonpharmacological treatment.

microbiota, the abundance of key species, and the reduction of BMI in NAFLD after nonpharmacological treatment.

## DISCUSSION

The present observational study on children's health corroborated the hypothesis of a connection between gut microbiota and NAFLD. The stool samples from children diagnosed with NAFLD, NASH, and NAFL, and HCs were analyzed using both 16S rRNA and metagenomic sequencing methods. The analysis aimed to determine the correlation between gut flora, NAFLD-related indicators, and metabolic pathways. The statistical results revealed that the abundance and composition of gut flora in patients with NAFLD were similar to those in HCs. The analysis pointed out that patients diagnosed with NAFLD, NASH, and NAFL, and HCs differed significantly in the bacterial taxa at the genus and species levels. The bacterial abundance had a strong positive correlation with NAFLD-related indexes and metabolic pathways. Notably, the study confirmed the important role of NAFLD-related bacteria in the advancement of the disease after nonpharmacological intervention. Hence, this study unveiled the link between gut microbiota and NAFLD and served as a scientific basis to elucidate the role of diverse bacterial taxa in disease advancement.

Using 16S rRNA analysis, we found that the microbial composition of patients with NAFLD and HCs was significantly different at the phylum, genus, and species levels, with slight differences in diversity between patients with NAFLD and HCs. The abundance of Fusobacteria and Proteobacteria was significantly higher in pediatric patients with NAFLD than in HCs at the phylum level, aligning with the findings of a previous study, which revealed an increase in the abundance of Proteobacteria in overweight patients with NAFLD (17). The upsurge in the abundance of Fusobacteria in pediatric patients with NAFLD was linked to polycellular infections in humans and animals (16). Additional discoveries included the over-representation of potential pathogens in pediatric patients with NAFLD, such as Enterobacteriaceae, *Escherichia-Shigella*, and *Fusobacteria*. Notably, these pathogenic species producing endotoxins could promote liver damage by activating macrophages via the lipopolysaccharide Toll-like receptor (TLR) 4 pathway (18). Our findings aligned with previous findings that established the damaging state of intestinal flora composition in pediatric patients with NAFLD.

We observed lower microbial diversity in patients with NASH and NAFL, as diagnosed using NAS and liver fibrosis stage by liver biopsy. The microbial diversity of patients with NASH was significantly higher than that of patients with NAFL, which might be associated with a higher abundance of multiple bacterial species. Several genera and species, such as *B. uniformis*, Lachnospiraceae bacterium 7_1_58FAA, *E. ventriosum*, *Roseburia*, and *Alistipes*, were significantly more abundant in patients with NASH compared with those with NAFL. In contrast, the abundance of Peptostreptococcaceae noname was lower in patients with NASH. NASH is the active form of NAFLD characterized by hepatic necroinflammation and rapid fibrosis progression (6). Dysbiosis and alterations in intestinal permeability through TLR signaling in hepatocytes induce inflammation progression from simple steatosis to NASH (19). The alterations in the gut microbiota have also been shown to contribute to the development of NAFL/NASH by mediating inflammation, insulin resistance, bile acid secretion, and choline metabolism, among others (20). The abundance of certain species of *Bacteroides*, such as *Alistipes*, which are highly bile tolerant and associated with high-fat diets, negatively correlated with obesity (21). The abundance of Lachnospiraceae bacterium 7_1_58FAA and *E. ventriosum*, both of which produce SCFAs, inversely correlated with body fat percentage (22). A majority of bacteria with a greater abundance in the feces of pediatric patients with NASH were beneficial bacteria. This finding suggested that the microbiome in patients with NASH might present a unique opportunity to interrupt disease progression.

This study demonstrated that the abundance of representative microbial taxa correlated with NAFLD-related indices. This finding suggested that the abundance of bacteria could be associated with NAFLD development. This study found a significant

correlation between the abundance of *R. intestinalis* and *R. inulinivorans* and GLU, AST, and ALT levels in NAFLD. A previous study demonstrated a correlation between the abundance of these species and glucose and lipid metabolism in NAFLD (23). Our finding of a positive correlation between TSH levels and the abundance of *Neisseria* unclassified corresponded to a previous study on patients with thyroid nodules, in which the relative abundance of *Neisseria* significantly increased, accompanied by inflammatory disorder (11). Thus, we theorized that these NAFLD-related bacteria might play a crucial role in NAFLD pathogenesis by affecting metabolism. Our study has identified a correlation between the abundance of previously unreported bacterial genera and species and the development of NAFLD in pediatric patients. This discovery provides a scientific foundation for understanding the role of bacteria in pediatric NAFLD and suggests that alterations in the abundance of bacterial genera and species could potentially serve as markers for disease remission.

The functional prediction analysis revealed the involvement of numerous bacterial taxa in pathways related to SCFA production, glucose metabolism, folate metabolism, and fatty acid metabolism. This finding suggested that the gut microbiota might affect the development of NAFL/NASH through energy utilization. The gut microbiota affected the energetic balance of the host by fermenting starch and nonstarter polysaccharides into SCFAs. This process, as previously shown, improved inflammation in NAFL/NASH (24). We identified key species, such as *Oscillibacter* unclassified, *D. longicatena*, *C. leptum*, *R. hominis*, *R. obeum*, *O. splanchnicus*, and *V. dispar*, that were closely related to acetate and butanoate synthesis. Moreover, these species were positively associated with pyruvate fermentation to acetate and lactate II. The abundance of *Clostridiales bacterium* 1_7_47FAA positively correlated with hexitol fermentation to lactate, formate, ethanol, and acetate.

Moreover, the abundance of Clostridiales bacterium 1_7_47FAA, *Ruminococcus gnavus*, and *C. freundii* showed a positive correlation with the TCA cycle VII, which was responsible for acetate production. In contrast, the abundance of *Blautia clarus* positively correlated with the fermentation of acetyl-CoA to butanoate II. *Oscillospira*, *Dorea*, *Clostridium*, *Roseburia*, *Odoribacter*, and *Ruminococcus* bacteria have been found to be capable of producing SCFAs (12, 25, 26). The reduced abundance of *Roseburia*, *Ruminococcus*, and *Clostridiales* in pediatric patients with NAFLD indicated their inability to metabolize SCFAs. SCFAs are regarded as a potential target for therapy in managing NAFLD, whereby modulating metabolic processes, preserving the integrity of the gut barrier, and restoring hepatic energy balance can effectively reduce hepatic inflammatory responses (27).

The interaction between SCFA metabolism and intestinal microbiota demonstrated that regulating SCFAs by microbiota had the potential to be a new strategy for treating NAFLD. The abundance of specific gut microbiota strongly correlated with various pathways of glucose metabolism. The abundance of *S. sanguinis*, *Enterococcus faecalis*, *Clostridium* sp. ATCC BAA 442, and *Blautia clarus*, for instance, positively correlated with the degradation of xylose IV. The coordination between gluconeogenesis and glycolysis was necessary for meeting energy needs under various conditions (28). The results showed that certain bacterial types regulated intestinal glucose metabolism while having the potential to produce SCFAs. The abundance of *W. confusa*, *A. odontolyticus*, *S. mitis*, *S. oralis*, *S. pneumoniae*, and *S. australis* positively correlated with N10-formyl-tetrahydrofolate biosynthesis and transformation of folate II. Reduced endogenous folate levels were observed in obese and diabetic patients, and this deficiency might be related to developing metabolic disorders such as NAFLD (29). Fat mainly accumulates in the liver through the esterification of free fatty acids (FFAs) derived from adipose tissue breakdown caused by insulin resistance in patients with NAFLD (30). In our study, the abundance of *C. freundii* and unclassified *Citrobacter* positively correlated with the fatty acid beta-oxidation pathway V and fatty acid salvage. During infection, host-derived fatty acids are an essential carbon source for pathogenic bacteria (31), and Citrobacter is known as an opportunistic pathogen found in various clinical human specimens

(32). We speculated that *Citrobacter*, under nonpathogenic conditions, could utilize FFAs for energy and reduce fatty acid accumulation in pediatric patients with NAFLD. These results illustrated that certain bacterial species in feces had a close relationship with metabolic pathways, and manipulating the gut microbiota composition might significantly impact human health via metabolism. Improving NAFLD progression by manipulating gut microbiota has great potential for future treatment.

The study employed a random forest algorithm to develop a classification model for identifying gut microbiome features in children diagnosed with NAFLD. Identifying such features might aid in the early assessment of NAFLD risk. The classification models based on an AUC value can help design personalized treatment and prevention strategies for NAFLD by facilitating microbiota modulation. The developed classification model had an AUC value of 0.969 for detecting NAFLD risk with a sensitivity and specificity of 0.914 and 0.949, respectively. This study identified gut microbiota features, including Lachnospiraceae ND3007, *Lachnoclostridium* of Lachnospiraceae, *Prevotella* and Prevotellaceae UCG-001 of Bacteroidaceae, *Ruminiclostridium* 6 of Ruminococcaceae, and *Coprococcus* 2 of *Coprococcus* in pediatric patients diagnosed with NAFLD.

This study involved a comparison of the abundance of NAFLD-related bacteria identified in the gut microbiota and the diversity of gut microflora before and after nonpharmacological treatment to assess the impact of altered BMI on an individual's microbial community. The abundance of gut microbiota species in pediatric patients diagnosed with NAFLD negatively correlated with the BMI after nonpharmacological treatment. This correlation suggested that the gut microflora diversity directly affected the alterations in BMI and might play a crucial role in such cases. Obvious differences in the abundance of NAFLD-related bacteria before and after treatment in children were observed. These differences correlated with the changes in BMI. After a treatment-induced reduction in BMI, the abundance of *B. longum*, *E. ventriosum*, Alcaligenaceae, Erysipelotrichaceae, *F. prausnitzii*, *Subdoligranulum* unclassified, *Lachnoclostridium*, *R. intestinalis*, *C. comes*, and *Bifidobacterium pseudocatenulatum* increased in pediatric patients with NAFLD. Conversely, the abundance of *Haemophilus, E. coli, Escherichia* unclassified, *Ruminococcus torques,* Lachnospiraceae UCG-008, and Lachnospiraceae bacterium 2_1_58FAA decreased. This study confirmed that the abundance of identified gut microbiota in pediatric patients with NAFLD was directly related to BMI. It also found that the bacterial abundance changed with the BMI. Korean adults with NAFLD had a significantly reduced abundance of gut species *Faecalibacterium, Coprococcus,* and Lachnospiraceae (33). In addition, Chinese pediatric patients with NAFLD and those with NASH had a significantly reduced abundance of *F. prausnitzii, Ruminiclostridium,* Bacteroidetes (*Alistipes*), *Lactobacillus, Oscillibacter,* and *Paraprevotella* species (9, 17). The study consistently found that pediatric patients with NAFLD had a decreased abundance of *F. prausnitzii*, *Lachnoclostridium*, and *Bifidobacterium*. However, the abundance increased significantly after the intervention. Based on previous research on intestinal flora in patients with NAFLD, *F. prausnitzii* and *Lachnoclostridium* were identified as butyrate-producing bacteria (34, 35). Furthermore, the study provided new insights into NAFLD-related bacteria in Chinese pediatric patients with NAFLD and identified several new marker species that were previously unknown, such as *E. ventriosum*, *Haemophilus*, *R. torques*, and *R. intestinalis*. These biomarkers might have a potential prognostic value, might predict overall mortality risk in pediatric patients with NAFLD, and might be useful as biological indicators in future clinical practice.

The present study provided evidence to support our hypothesis that an association existed between gut microbiota and NAFLD. We found that the variations in gut microbiota might potentially play a crucial role in managing NAFLD compared with a single species. Furthermore, we identified several NAFLD-related bacteria, including *Lachnoclostridium*, *Escherichia-Shigella*, and *F. prausnitzii*, whose abundance positively correlated with NAFLD-specific indicators and metabolic pathways. These results revealed new aspects of gut microbiota in Chinese children and adolescents with NAFLD, mainly NASH, and established a microbiota diagnostic profile for NAFLD,

opening up therapeutic avenues for probiotics/prebiotics. However, the sample size in this study was relatively small. Hence, further investigations with larger, well-characterized cohorts are needed to better describe the correlation between intestinal flora and NAFL/NASH.

## ACKNOWLEDGMENTS

This study was supported by the Digestive Medical Coordinated Development Center of Beijing Municipal Administration of Hospitals (No. XXT22), Beijing Natural Science Foundation (No. 7222060), Natural Science Foundation of Liaoning Province (No. 2020JH2/10300041), Capital Health Development Research Project (Shoufa 2020-1-2024), Beijing Hospitals Authority's Ascent Plan (No. DFL20221003), Beijing Hope Run Special Fund of Cancer Foundation of China (No. LC2021A06), and Beijing Natural Science Foundation (No. 7222153).

J.Z. conceptualized the study, led the investigation, and acquired funding. M.S. curated the data and wrote the original draft. C.Z. was involved in the investigation and helped with project administration. G.L. contributed to data curation, formal analysis, and visualization, and reviewed and edited the manuscript. C.L., X.G., and C.P. contributed to data analysis and visualization. Y.K., D.L., W.Y., B.C., L.F., Y.Y., J.W., J.Z., Y.F., and X.M. were involved in the investigation and provided clinic resources. Y.L. helped with the methodology and experiment resources. L.W. proposed the methodology, supervised the study, and reviewed and edited the manuscript.

## AUTHOR AFFILIATIONS

[1]National Center for Children's Health, Beijing Children's Hospital, Capital Medical University, Beijing, China
[2]CAS Key Laboratory of Microbial Physiological and Metabolic Engineering, State Key Laboratory of Microbial Resources, Institute of Microbiology, Chinese Academy of Sciences, Beijing, China
[3]Department of Scientific Research, Microvita Medical Technology Co., Beijing, China
[4]Microbial Resources and Big Data Center, Institute of Microbiology, Chinese Academy of Sciences, Beijing, China
[5]Department of Thoracic Surgery, National Cancer Center/National Clinical Research Center for Cancer/Cancer Hospital, Chinese Academy of Medical Sciences and Peking Union Medical College, Beijing, China

## AUTHOR ORCIDs

Jing Zhang ⓘ http://orcid.org/0000-0002-2832-239X
Mengxuan Shi ⓘ http://orcid.org/0009-0001-7971-3753
Guangcai Liang ⓘ http://orcid.org/0000-0002-4767-7698
Liming Wang ⓘ http://orcid.org/0009-0008-9180-0981

## FUNDING

| Funder | Grant(s) | Author(s) |
| --- | --- | --- |
| Digestive Medical Coordinated Development Center of Beijing Hospitals Authority | XXT22 | Jing Zhang |
| Beijing Natural Science Foundation | 7222060 | Jing Zhang |
| Liaoning Natural Science Foundation | 2020JH2/10300041 | Jie Wu |
| Beijing Hospitals Authority's Ascent Plan | DFL20221003 | Jie Wu |
| Beijing Hope Run Special Fund of Cancer Foundation of China | LC2021A06 | Yong Li |
| Beijing Natural Science Foundation | 7222153 | Yong Li |

| Funder | Grant(s) | Author(s) |
|---|---|---|
| Capital Health Development Research Project | Shoufa 2020-1-2024 | Jing Zhang |

## AUTHOR CONTRIBUTIONS

Jing Zhang, Conceptualization, Funding acquisition, Investigation | Mengxuan Shi, Writing – original draft, Data curation | Chunna Zhao, Investigation, Project administration | Guangcai Liang, Data curation, Formal analysis, Visualization, Writing – review and editing | Chuan Li, Data curation, Visualization | Xiaomeng Ge, Visualization, Data curation | Caixia Pei, Data curation, Visualization | Yawei Kong, Investigation, Resources | Dongdan Li, Investigation, Resources | Wenli Yang, Investigation, Resources | Bingyan Cao, Investigation, Resources | Libing Fu, Investigation, Resources | Yinkun Yan, Investigation, Resources | Jie Wu, Investigation, Resources | Jin Zhou, Investigation, Resources | Yongli Fang, Investigation, Resources | Xi Meng, Investigation, Resources | Yong Li, Resources, Methodology | Liming Wang, Methodology, Writing – review and editing, Supervision

## DATA AVAILABILITY

Sequences analyzed in this study are available in GenBank with the accession number PRJNA914785.

## ETHICS APPROVAL

This study was granted ethical clearance by the Human Ethics Committee of Beijing Children's Hospital, Capital Medical University. All participants provided written parental informed consent, and written child assent was also secured.

## ADDITIONAL FILES

The following material is available online.

Open Peer Review

**PEER REVIEW HISTORY (review-history.pdf).** An accounting of the reviewer comments and feedback.

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
