## [Reviewer comments · Microbiology Spectrum]

Microbiology Spectrum

Role of intestinal flora in the development of children with nonalcoholic fatty liver disease

Jing Zhang, Chunna Zhao, Mengxuan Shi, Dongdan Li, Yawei Kong, Wenli Yang, Bingyan Cao, Libing Fu, Yinkun Yan, Jie Wu, Jin Zhou, Yongli Fang, Xi Meng, Yong Li, Chuan Li, Caixia Pei, Xiaomeng Ge, Guangcai Liang, and Liming Wang

Corresponding Author(s): Liming Wang, Institute of Microbiology Chinese Academy of Sciences

Review Timeline:

Submission Date:	March 19, 2023
Editorial Decision:	April 10, 2023
Revision Received:	July 26, 2023
Editorial Decision:	August 13, 2023
Revision Received:	October 3, 2023
Editorial Decision:	October 9, 2023
Revision Received:	October 24, 2023
Accepted:	November 11, 2023

Editor: Wei-Hua Chen

Reviewer(s): Disclosure of reviewer identity is with reference to reviewer comments included in decision letter(s). The following individuals involved in review of your submission have agreed to reveal their identity: Wei-Kai Wu (Reviewer #2)

Transaction Report:

DOI: <https://doi.org/10.1128/spectrum.01006-23>

April 10, 2023

Dr. Liming Wang
Institute of Microbiology Chinese Academy of Sciences
Beijing
China

Re: Spectrum01006-23 (Role of intestinal flora in the development of children with nonalcoholic fatty liver disease)

Dear Dr. Liming Wang:

Link Not Available

Sincerely,

Wei-Hua Chen

Journals Department
Reviewer comments:

Reviewer #1 (Comments for the Author):

Zhang et al. utilized a 16S rRNA and metagenomic sequencing approach on fecal samples from pediatric patients with NAFLD, NASH, and AFL, as well as HCs, to investigate the correlation among NAFLD-related indices, metabolic pathways, and gut microbiota. Overall, the topic is interesting and important for the field. However, there are some questions for the authors, which are included below.

In line 91-92, the authors mentioned, "The information of the participants, including sex, birth date, height, and weight, were

obtained from their parents." I wonder if this information is corrected or standardized (e.g., different families might use different scales, and some participants might weigh themselves after meals while others do not).

In line 190-191, the authors mentioned, "The BMI, fasting blood glucose, uric acid, TSH, and two liver enzyme indices were measured." However, in Table 1, the HC group did not show fasting blood glucose, uric acid, TSH, ALT, and AST.

Why did the authors only show 13 samples in the Table (NAFL = 5, NASH = 8)? Additionally, the significance of age between the NAFL and NASH groups was significantly different ($P = 0.007$).

In Figures 1 and 2, the authors showed HC and NAFLD groups, but in Figure 3, the authors showed three groups, which may confuse readers. Please clarify the three groups at first.

For Figure 2C, please improve the figure quality, as the words in the figures were too small in the current version.

For Figure 3, the samples were different from the others (HC, $n = 35$, NAFL, $n = 5$, NASH, $n = 8$). Did the authors select the samples and try to show the helpful data for their hypothesis?

For Figure 4, the same questions as comment 6.

For Figure 8, please use scientific terms to explain the BMI increased and decreased.

Table 3 is not suitable for the article and may be more suitable for a review. I suggest the authors delete Table 3 and add the information in the "Discussion" section.

Please carefully check the whole manuscript when submitting the revision to avoid some Chinese words (e.g., line 303).

Reviewer #2 (Comments for the Author):

Zhang et al. conducted an observational study to investigate the gut microbial features of pediatric NAFLD ($n=79$) when comparing with a healthy control (HC) group ($n=35$). The authors used both 16S rRNA V3-V4 amplicon and shotgun metagenome sequencing for the gut microbiome analysis and concluded *Faecalibacterium prausnitzii*, Ruminococcaceae and Bifidobacteriaceae were decreased in NAFLD children whereas *Lachnospiraceae*, *Escherichia-Shigella*, *Ruminococcus gnavus* group and *Lachnospiraceae* were increased. The authors also built a Random Forest prediction model with gut microbial features which is claimed to distinguish NAFLD from HC in children with an AUROC of 0.969. Besides, a small proportion of NAFLD patients ($n=15$?) received nonmedical treatment (lifestyle modification) and noted some gut bacteria were altered after the intervention as well. The intent of this study to discover gut microbial feature in NAFLD children is good, however, may require a more mindful and sophisticated analytical approach to achieve the goal. There were several concerns needing to be addressed which are as follows:

1. In table 1, the authors showed that the average BMI in NAFLD children (28.37kg/m^2) is significantly higher than HC (17.63kg/m^2). However, the table showed the average weight in NAFLD (72.38) is lower than HC (86.79kg), and the average height in NAFLD (158.22cm) is higher than HC (102.95cm). It means the NAFLD patients have higher BMI, but lower body weight and higher height. Mathematically, it does not make sense.
2. In the materials and methods, the definition of NAFLD activity score (NAS) is COMPLETELY WRONG (line 104-107). What is mentioned in line 105-107 is called NASH fibrosis stage, not NAS. This obvious mistake of NAFLD knowledge may raise the concern of professionalism and data reliability.
3. For the microbiome analysis, there are several contradictory descriptions which may raise the concern whether the authors interpretate the data correctly. For example, in line 227-229: "The statistical models showed that the abundance and composition of flora species in patients with NAFLD and HCs were similar ($P < 0.05$)." Here, the authors stated the composition of gut microbiota between NAFLD and HCs were similar but highlighted a $P < 0.05$ for the PCoA data. In line 230-232: "As shown in Figure 2C and D, the observed species did not differ significantly by sex in patients with NAFLD and HCs ($P < 0.05$)." Here, again, the authors compare the alpha diversity in male and female among NAFLD and HC subjects and stated no significant difference but having a $P < 0.05$! There are still many errors like this in the whole manuscript.
4. For all box plots in the manuscript, all the labels are too small and unclear for both readers and reviewers.
5. In line 270, the figure legend showed "Genera with significant differences in abundance between patients with NAFLD and HCs (HC, $n = 35$, NAFLD, $n = 79$, $P < 0.05$, $P < 0.01$)." It is weird to see two P values here without any explanation, and many like this in the manuscript.
6. In Figure 6, what does the authors mean for NAFLD-related metabolic pathway? It appears to me that the authors simply

correlate the microbial taxa (from V3-V4 amplicon or MAG??) with the annotated genes from shotgun metagenome of fecal samples among the 79 NAFLD patients. I cannot see any insightful meaning for the correlation analysis. To investigate the NAFLD-related metabolic pathway, the authors should conduct RNA-seq or proteomics analysis for the biopsied liver tissues.

7. For Figure 7, the concern of overfitting in the prediction modelling exists since no validation cohort is applied.

8. For Figure 8, it looks like that only 7 subjects received the nonmedical treatment, however, the authors stated they had 15 children with NAFLD to receive the intervention (line 93-95).

9. Is there any analysis of gut microbiome comparison between NAFLD and HC by using shotgun metagenome? It seems to me that the differential abundance between NAFLD and HC has only been demonstrated by 16S rRNA amplicon sequencing in the manuscript and figures.

10. Finally, I suggest the manuscript to have professional English editing.

Reviewer #3 (Comments for the Author):

In this manuscript, the authors explored the role of intestinal flora in the development of children with NAFLD and made comparisons between the gut microbiota in NAFL and that in NASH. A considerable work was done in this article, however, there are some parts/issues that remains to be more clearly explained. I suggest the following questions be taken into account:

1. This article is mainly about gut microbiome and NAFLD, therefore, the previous research background should be carefully described, however, in the introduction part, only few lines were mainly about the related research background, especially literature about gut microbiome and NAFL/NASH. And in line 69, the word "several" is not accurate, that even contradicts with those many articles the authors listed in the Table 3.
2. 79 patients with NAFLD were included in this study, of whom 8 had NASH and 5 had NAFL based on NAS. So, what categories do the rest 66 NAFLD patients belong to? And their NAS? The authors should clarify.
3. In Table 1, the NAS and the seven biochemical data in the NC ground are blank. I suggest these data be supplemented. After that, for example, you can have a map for the control group in Figure 5, right?
4. Several bioinformatic software packages (e.g QIIME, version 1.9.1) adopted in this study were not up-to-date, and that will influence the accuracy and reliability of the analysis results in a certain degree.
5. Line 132-134, why did not the authors use the RDP classifier with the corresponding RDP reference dataset or use the Greengenes classifier with the corresponding Greengenes reference dataset?
6. Line 138-140, QIIME and LEfSe are two difference packages, therefore, QIIME can not be used to analyze the effect size (LEfSe). In addition, LEfSe is the abbreviation of "LDA Effect Size", therefore in this sentence, there is no need to mention LDA.
7. Both 16s rRNA and Metagenomic sequencing were performed in this study, it seems that the authors did not clarify which data were performed by 16S and which by Metagenomic or if they performed 16S and metagenomic seq on all the samples and why?
8. The authors should provide more details about the machine learning method and results, for instance, the data they selected, the confusion matrix of the results and how did they train and test those data.

Staff Comments:

Preparing Revision Guidelines

Please return the manuscript within 60 days; if you cannot complete the modification within this time period, please contact me. If you do not wish to modify the manuscript and prefer to submit it to another journal, please notify me of your decision immediately so that the manuscript may be formally withdrawn from consideration by Microbiology Spectrum.

Response to Reviewer 1 Comments

Point 1: In line 91-92, the authors mentioned, "The information of the participants, including sex, birth date, height, and weight, were obtained from their parents." I wonder if this information is corrected or standardized (e.g., different families might use different scales, and some participants might weigh themselves after meals while others do not).

Response 1: Thank you for addressing the reviewer's question. We concur with your perspective. We conducted professional training sessions to educate parents on how to measure their child's height and weight. We instructed parents to ensure that the children wore consistent clothing half an hour before meals and measured them with a standardized weight and height gauge. As a result, the information collected from the participants' parents is standardized.

Point 2: In line 190-191, the authors mentioned, "The BMI, fasting blood glucose, uric acid, TSH, and two liver enzyme indices were measured." However, in Table 1, the HC group did not show fasting blood glucose, uric acid, TSH, ALT, and AST.

Response 2: Thank you for addressing the reviewer's question. We concur with your perspective. This study focused on comparing only the differences in height, weight, and BMI between healthy control (HC) and non-alcoholic fatty liver disease (NAFLD) children. To investigate the differences between non-alcoholic steatohepatitis (NASH) and NAFL children, the data collected included height, weight, BMI, fasting blood glucose, uric acid, thyroid-stimulating hormone (TSH), alanine transaminase (ALT), and aspartate transaminase (AST). The table's content and the results' description have been revised and are available on page 11, lines 250-253, and Table 2 respectively.

Point 3: Why did the authors only show 13 samples in the Table (NAFL = 5, NASH = 8)? Additionally, the significance of age between the NAFL and NASH groups was significantly different ($P = 0.007$).

Response 3: Thank you for addressing the reviewer's question. Out of the 79 children with non-alcoholic fatty liver disease (NAFLD), only 13 children whose parents provided consent for liver puncture underwent in our analysis. We did not make any manual intervention or data selecting. Following non-alcoholic steatohepatitis (NASH) activity score (NAS) evaluation, we identified eight cases of NASH and five cases of non-alcoholic fatty liver (NAFL), and conducted further analysis with these 13 samples. We conducted statistical analysis on the data using SPSS 20.0 and obtained a P-value of 0.023. The findings have been added and revised in the manuscript's Table 3, located on page 13-14, lines 279-300.

Point 4: In Figures 1 and 2, the authors showed HC and NAFLD groups, but in Figure 3, the authors showed three groups, which may confuse readers. Please clarify the three groups at first.

Response 4: We would like to express our sincere gratitude for the reviewer's suggestion. The presentation of the results has been revised in the updated version of the manuscript. Figures 2 and 3 illustrate the variations in intestinal flora diversity and species abundance between healthy control (HC)

children and those with non-alcoholic fatty liver disease (NAFLD). Figure 4 identifies the differences in intestinal flora diversity and species abundance between children with non-alcoholic steatohepatitis (NASH) and NAFL, and supports the findings obtained from the liver puncture pathology evaluation. These updated and supplemented results and figures are now available on pages 12-15 as Figures 2-4 in the manuscript.

Point 5: For Figure 2C, please improve the figure quality, as the words in the figures were too small in the current version.

Response 5: Dear Reviewer, thank you for your inquiry. We concur with your standpoint. In this study, the analysis of gender-related gut flora in children with HC and NAFLD revealed no significant differences. Therefore, we deemed it necessary to exclude this section from the revised version of our manuscript. Figures 1-8 have been updated with clearer images. The font size of the figures has also been enlarged, and the picture pixel has been increased for better clarity and improved readability.

Point 6: For Figure 3, the samples were different from the others (HC, n = 35, NAFL, n = 5, NASH, n = 8). Did the authors select the samples and try to show the helpful data for their hypothesis?

Response 6: We would like to express our genuine gratitude to the reviewer for their valuable suggestion. Out of 79 children with NAFLD, only those who agreed to undergo liver puncture were selected. After NAS score evaluation, we identified eight cases of NASH and five cases of NAFL and conducted further analysis on these 13 samples. In the revised manuscript, we have modified the representation of Figure 4 to emphasize the variations in gut flora diversity and species abundance between children with NASH and NAFL. The updated illustration has been included in the manuscript (Page 15, Figure 4).

Point 7: For Figure 4, the same questions as comment 6.

Response 7: We are grateful for the valuable suggestion provided by the reviewer. Among the 79 children diagnosed with NAFLD, only those who provided consent for liver puncture were included in our analysis. After evaluating the NAS score, we identified eight cases of NASH and five cases of NAFL. We proceeded with further analysis on these 13 samples. In the revised version of the manuscript, we made modifications Figure 4 to highlight the variations in gut flora diversity and species abundance among NASH and NAFL afflicted children. These modifications have been incorporated in the manuscript (Page 15, Figure 4) for better comprehension.

Point 8: For Figure 8, please use scientific terms to explain the BMI increased and decreased.

Response 8: We are grateful for the invaluable input provided by the reviewer. We conducted an analysis of BMI changes in 15 children with NAFLD who consented to follow-up appointments. Our findings indicate that 5 of the participants exhibited a decrease in BMI exceeding 5% after non-pharmacological treatment and were identified as effective, 2 were observed to have an increase in BMI exceeding 5% and were identified as ineffective, while the other 8 showed a change in BMI

below 5% and were deemed to have no effect. We have incorporated these results in the manuscript (Page 20, Line 408-414) for better comprehension.

Point 9: Table 3 is not suitable for the article and may be more suitable for a review. I suggest the authors delete Table 3 and add the information in the "Discussion" section.

Response 9: Thank you for the reviewer's inquiry. We have relocated the contents of Table 3 to the "Introduction" section, and it is now referred to as Table 1. Additionally, we included the present state of domestic and international studies that associate intestinal flora with NAFLD/NASH interactions within the "Introduction" section. Furthermore, we have introduced the discrepancies between the findings of this examination and those of national and international reports in the "Discussion" section. The changes mentioned above have been implemented and edited in the manuscript (Page 3-4, Line 77-81, Table 1).

Point 10: Please carefully check the whole manuscript when submitting the revision to avoid some Chinese words (e.g., line 303).

Response 10: We appreciate your comments. We sincerely apologize for our carelessness. We have thoroughly revised the manuscript, removing all Chinese words and correcting spelling, typing and grammatical errors. The changes have been applied across the manuscript (Page 17, Figure 5).

Response to Reviewer 2 Comments

Point 1: In table 1, the authors showed that the average BMI in NAFLD children (28.37kg/m^2) is significantly higher than HC (17.63 kg/m^2). However, the table showed the average weight in NAFLD (72.38) is lower than HC (86.79 kg), and the average height in NAFLD (158.22 cm) is higher than HC (102.95 cm). It means the NAFLD patients have higher BMI, but lower body weight and higher height. Mathematically, it does not make sense.

Response 1: Thank you for your inquiry. We concur with your observations. Within the revised version of the manuscript, we have included the accurate data results for height (151.28 ± 14.87) and weight (40.14 ± 12.11) and BMI (17.17 ± 2.34) of HC children, and height (158.22 ± 14.37) and weight (72.38 ± 20.01) and BMI (28.37 ± 4.36) of NAFLD children. We conducted a statistical analysis of the BMI values for children with HC and NAFLD utilizing SPSS 20.0. Our findings indicated a significant difference between the two groups, with a P-value of 0.001. We have implemented the aforementioned changes in the manuscript (Page 11, Line 250-253, Table 3).

Point 2: In the materials and methods, the definition of NAFLD activity score (NAS) is COMPLETELY WRONG (line 104-107). What is mentioned in line 105-107 is called NASH fibrosis stage, not NAS. This obvious mistake of NAFLD knowledge may raise the concern of professionalism and data reliability.

Response 2: Thanks for the reviewer's question. We agree with your opinion. We understand the importance of accurate reporting and maintaining professional standards in our research. We revised the section on describe the NAFLD activity score (NAS) and liver fibrosis stage accurately in the revised manuscript. The revised manuscript includes all these updates and modifications (Page 6-7, Line 127-144, and Figure 1).

Point 3: For the microbiome analysis, there are several contradictory descriptions which may raise the concern whether the author's interpretate the data correctly. For example, in line 227-229: "The statistical models showed that the abundance and composition of flora species in patients with NAFLD and HCs were similar ($P < 0.05$)." Here, the authors stated the composition of gut microbiota between NAFLD and HCs were similar but highlighted a $P < 0.05$ for the PCoA data. In line 230-232: "As shown in Figure 2C and D, the observed species did not differ significantly by sex in patients with NAFLD and HCs ($P < 0.05$)." Here, again, the authors compare the alpha diversity in male and female among NAFLD and HC subjects and stated no significant difference but having a $P < 0.05$! There are still many errors like this in the whole manuscript.

Response 3: Thank you for the reviewer's inquiry. We sincerely apologize for our carelessness. We have updated the method of annotating prominence in the manuscript as follows:

1. Line 266-272, the statements of "The statistical models showed that the abundance and composition of flora species in patients with NAFLD and HCs were similar ($P < 0.05$). Next, the intestinal flora in male and female patients with NAFLD and HCs showed that sex did not affect the microbiota compositions. As shown in Figure 2C and D, the observed species did not differ significantly by sex in

patients with NAFLD and HCs ($P < 0.05$.)” were corrected as “No significant differences in species richness and evenness were observed between HCs and patients with NAFLD in median alpha diversity (observed species, Chao1, Shannon, and Simpson diversity indices) (Fig. 1C). However, the observed species in patients with NAFLD were more dispersed than those observed in HCs. A PCoA analysis of the gut microbiota samples of the children is presented in Figure 1D. The statistical models revealed that the composition and abundance of flora species in patients with NAFLD and HCs were similar.”

Point 4: For all box plots in the manuscript, all the labels are too small and unclear for both readers and reviewers.

Response 4: Thank you for your inquiry. We concur with your observations. We have updated all figures in the manuscript with clearer images, enlarging the font and improving picture pixelation (Figure 1-8).

Point 5: In line 270, the figure legend showed "Genera with significant differences in abundance between patients with NAFLD and HCs (HC, n = 35, NAFLD, n = 79, $P < 0.05$, $P < 0.01$)." It is weird to see two P values here without any explanation, and many like this in the manuscript.

Response 5: Thank you for your inquiry. We acknowledge and concur with your perspective. In response to your feedback, we have made alterations to the annotation method employed in our manuscript to enhance the identification of prominence:

Line 291-292, the statements of “Genera with significant differences in abundance between patients with NAFLD and HCs (HC, n = 35, NAFLD, n = 79, $P < 0.05$, $P < 0.01$.)” were corrected as “Multiple bacterial species with significantly different abundance were found in the intestinal flora of patients with NAFLD and HCs (HC, n = 35, NAFLD, n = 79, $P < 0.05$.)”

Point 6: In Figure 6, what does the authors mean for NAFLD-related metabolic pathway? It appears to me that the authors simply correlate the microbial taxa (from V3-V4 amplicon or MAG??) with the annotated genes from shotgun metagenome of fecal samples among the 79 NAFLD patients. I cannot see any insightful meaning for the correlation analysis. To investigate the NAFLD-related metabolic pathway, the authors should conduct RNA-seq or proteomics analysis for the biopsied liver tissues.

Response 6: We appreciate the valuable suggestion provided by the reviewer. The present study analyzed the correlation between strain and metabolic pathway through metagenomic sequencing of fecal samples from 79 children suffering from NAFLD. By conducting metagenomic sequencing, we were able to obtain genetic data that facilitated the analysis of metabolic pathways associated with strains that were significantly altered in NAFLD. In the revised manuscript, we have modified the terminology used to describe the metabolic pathways associated with NAFLD. Specifically, we now refer to them as metabolic pathways that exhibit a significant association with NAFLD-associated strains, with an aim to enhance the precision of our language.

Point 7: For Figure 7, the concern of overfitting in the prediction modelling exists since no validation cohort is applied.

Response 7: We appreciate your feedback and constructive comments on our manuscript. We agree that model validation is a vital step in ensuring the ability of our model to generalize to new data and avoiding overfitting. In response to your concern, we have made updates to our manuscript to address this issue. We have incorporated a more detailed and comprehensive presentation of the machine learning approach employed in our revised study (Page 8-9, Line 187-198). Specifically, the Random Forest model was trained with a total of 114 stool samples, 35 of which were healthy and the remaining 79 were NAFLD samples. For the data training process, two-thirds of all data were allocated as the training set, while the remaining third was kept as the test set. To mitigate the influence of confounding variables from correlated data, we normalized the filtered relative abundances. We express our gratitude for your insightful feedback and inputs.

Point 8: For Figure 8, it looks like that only 7 subjects received the nonmedical treatment, however, the authors stated they had 15 children with NAFLD to receive the intervention (line 93-95).

Response 8: We genuinely appreciate the insightful suggestion provided by the reviewer and concur with your point of view. The present study monitored the BMI changes over time in 15 children with NAFLD who voluntarily participated in the follow-up protocol. Our analysis showed that 5 of them achieved a reduction in their BMI exceeding 5% following non-pharmacological treatment, which we deemed effective. In contrast, 2 of them experienced a BMI increase greater than 5% and were identified as ineffective, while the remaining 8 showed less than 5% variation in BMI and were categorized as having no effect. Based on these outcomes, we conducted additional analyses of samples with BMI changes exceeding 5%, which have been included in the revised manuscript on Page 20, Line 408-414.

Point 9: Is there any analysis of gut microbiome comparison between NAFLD and HC by using shotgun metagenome? It seems to me that the differential abundance between NAFLD and HC has only been demonstrated by 16S rRNA amplicon sequencing in the manuscript and figures.

Response 9: Thank you for the valuable question raised by the reviewer, and we concur with your perspective. The present study included comparative analysis of the intestinal microbiota in a group of 35 healthy children (HC) and 79 children diagnosed with NAFLD, employing 16s rRNA sequencing of stool samples. Furthermore, we conducted metagenomic sequencing of the intestinal bacterial flora in 8 NASH and 5 NAFL children stool samples as part of the comparative flora study. We have included additional information regarding this analysis in the revised manuscript.

Point 10: Finally, I suggest the manuscript to have professional English editing.

Response 10: Thank you for providing your feedback on our paper. We appreciate your insights and concur with your opinion. In response to your comments, we have modified certain phrases in the manuscript and enlisted the assistance of native English-speaking professionals to edit and revise the language throughout the manuscript.

Response to Reviewer 3 Comments

Point 1: This article is mainly about gut microbiome and NAFLD, therefore, the previous research background should be carefully described, however, in the introduction part, only few lines were mainly about the related research background, especially literature about gut microbiome and NAFL/NASH. And in line 69, the word "several" is not accurate, that even contradicts with those many articles the authors listed in the Table 3.

Response 1: We are grateful to the reviewer for raising a pertinent question and for the opportunity to address it. We concur with your opinion and have taken the following actions in response: In the revised manuscript, we have provided updated information about domestic and international studies on the interaction between intestinal microbiota and NAFLD/NASH in the "Introduction" section. We have included Table 1 for the summary of the literature. Additionally, we have rectified any inaccurate English descriptions throughout the article. To ensure greater clarity and precision, we have removed vague words such as "several." The revised manuscript includes all these updates and modifications (Page 3-4, Line 77-81, Table 1).

Point 2: 79 patients with NAFLD were included in this study, of whom 8 had NASH and 5 had NAFL based on NAS. So, what categories do the rest 66 NAFLD patients belong to? And their NAS? The authors should clarify.

Response 2: We appreciate the reviewer's question and agree with their suggestion. Out of the 79 children diagnosed with NAFLD in this study, only those who consented to undergo liver puncture were included in our analysis. Based on the evaluation of the NAFLD activity score (NAS), we identified eight cases of NASH and five cases of NAFL and performed further analyses on these 13 samples. We have revised and supplemented this information in the revised manuscript (Page 6-7, Line 127-144, and Figure 1, Page 13, Line 295-300).

Point 3: In Table 1, the NAS and the seven biochemical data in the NC ground are blank. I suggest these data be supplemented. After that, for example, you can have a map for the control group in Figure 5, right?

Response 3: In response to the reviewer's inquiry, we would like to express our sincere apologies for our carelessness. In this study, we solely investigated the differences in height, weight, and BMI between children with HC and NAFLD. Meanwhile, the data including height, weight, BMI, fasting blood glucose, uric acid, TSH, ALT, and AST were employed to compare the distinctions between children with NASH and NAFL. The findings displayed in Figure 5 portrayed the gut flora compositions of children with NAFLD that correlated with their respective NAFLD indicators, utilizing the metagenomic sequencing data of 79 from the NAFLD group. The contents of the table and the presentation of the results have been adjusted in the manuscript accordingly (Page 11, Line 250-253, Table 3).

Point 4: Several bioinformatic software packages (e.g QIIME, version 1.9.1) adopted in this study were

not up-to-date, and that will influence the accuracy and reliability of the analysis results in a certain degree.

Response 4: Thanks for the reviewer's question. We appreciate the reviewer's comment regarding the software used in this study. After reexamining our data processing workflow, we found that we had actually used version 2.0 in our study. We have corrected this information in our manuscript and thank you for bringing it to our attention. Additionally, we did use QIIME version 2.0 in our study, and we were attentive to making sure that all software packages used were up-to-date and aligned with current best practices in our field. Nevertheless, we will strive to use the most current versions of software packages in future studies available to ensure we are producing the most accurate and reliable results possible.

Point 5: Line 132-134, why did not the authors use the RDP classifier with the corresponding RDP reference dataset or use the Greengenes classifier with the corresponding Greengenes reference dataset?

Response 5: Thanks for the reviewer's question. We appreciate the reviewer's comment. We chose to use the Mothur program to process the raw sequencing data and perform quality control according to previously published protocols, and classify taxonomy based on the SILVA database. The classifiers have been chosen based upon its proven accuracy and reliability in analyzing microbiome data, as well as our previous experience with these tools. We have corrected this information in our manuscript and thank you for bringing it to our attention. We have revised and supplemented this information in the revised manuscript (Page 8, Line 174-176).

Point 6: Line 138-140, QIIME and LEfSe are two different packages, therefore, QIIME can not be used to analyze the effect size (LEfSe). In addition, LEfSe is the abbreviation of "LDA Effect Size", therefore in this sentence, there is no need to mention LDA.

Response 6: We express our sincere appreciation for the helpful suggestion from the reviewer. We wholeheartedly concur with your opinion and have made the necessary adjustments to rectify the imprecise description:

Line 183-186, the statements of "Furthermore, QIIME was used to analyze the alpha diversity (observed species, Chao1, Shannon, and Simpson), beta diversity [principal coordinate analysis (PCoA)], linear discriminant analysis (LDA), and effect size (LEfSe) [20]." were corrected as "Furthermore, QIIME was used to analyze the alpha diversity (observed species, Chao1, Shannon, and Simpson) and beta diversity [principal coordinate analysis (PCoA)]. LEfSe was applied to compare significant differences in taxa between patients with NAFLD and HCs using a *P* value <0.05".

Point 7: Both 16s rRNA and Metagenomic sequencing were performed in this study, it seems that the authors did not clarify which data were performed by 16S and which by Metagenomic or if they performed 16S and metagenomic seq on all the samples and why?

Response 7: We are truly grateful for the reviewer's valuable suggestion. This study enrolled a total of 35 children with HC and 79 children with NAFLD, and comparative flora assessments were conducted

through sequencing the 16s rRNA of their stool samples. Moreover, in order to create a prediction model for disease and validate the population cohort, the sample data underwent 16s rRNA sequencing. As for the 8 NASH and 5 NAFL children stool samples, the intestinal flora were sequenced utilizing metagenomic sequencing for comparative flora study and correlation analysis of strains with clinical indicators of NAFLD and metabolic pathways. We resorted to distinct sequencing methods depending on the fecal sample DNA requirements and applied the appropriate sequencing technique to each distinct sample. Pages 7, 8 and 9 respectively contain additional content detailing this information (Page 8, Line 168-171, Page 9, Line 200-201, Page 11, Line 256-258, Page 14, Line 305-307).

Point 8: The authors should provide more details about the machine learning method and results, for instance, the data they selected, the confusion matrix of the results and how did they train and test those data.

Response 8: We sincerely appreciate the reviewer's suggestion. We agree with your opinion. We have revised our manuscript to incorporate a more detailed and comprehensive presentation of the machine learning approach employed in our study. Specifically, we have provided additional information regarding the dataset including the source and size. In addition, we have supplied a detailed account of the confusion matrix of the results. We believe that these modifications enhance the overall clarity and quality of our writing as well as provide our readers with a deeper understanding of our research. We have revised and supplemented this information in the revised manuscript (Page 8-9, Line 187-198). Thank you for your valuable feedback, which has contributed to the improvement of our manuscript.

August 13, 2023

Dr. Liming Wang
Institute of Microbiology Chinese Academy of Sciences
No.1 West Beichen Road, Chaoyang District, Beijing, China 100101
Beijing
China

Re: Spectrum01006-23R1 (Role of intestinal flora in the development of children with nonalcoholic fatty liver disease)

I hope this email finds you well. Firstly, I extend my sincere apologies for the delay in reaching a decision on your revised manuscript. The summer holiday period contributed to this unforeseen delay.

I want to express my gratitude for your submission to Microbiology Spectrum. Your revised manuscript has undergone rigorous evaluation by two experts in the field. While one reviewer expressed favorable opinions regarding your results, the other raised concerns relating to the number of samples sequenced and the biological significance of correlations between microbial abundance and metabolic pathways.

Addressing the first concern, I conducted a thorough check on the NCBI bioproject referenced in your manuscript. It has come to my attention that there was a misunderstanding regarding the sample count. In reality, a total of 201 samples were sequenced, as opposed to the 13 mentioned by the reviewer (for more information, please refer to <https://www.ncbi.nlm.nih.gov/bioproject/PRJNA914785>). I believe that this misunderstanding stemmed from the way the results were presented, particularly the initial statement at line 140, which read "Eight participants had NASH, and five had NAFL in the NAFLD group." I kindly request that you revise the manuscript to rectify these potentially misleading statements.

Furthermore, I wish to bring to your attention the repeated use of "bacterial strains" throughout the manuscript. Given that the study's 16S sequencing accurately identifies bacteria only at the genus level, this terminology is inaccurate and could be misleading. I kindly ask you to address this issue and use accurate terminology.

Please note, this is going to be the last round of revision.

Link Not Available

Sincerely,

Wei-Hua Chen

Journals Department
Reviewer comments:

Reviewer #1 (Comments for the Author):

The authors carefully addressed my previous questions and concerns.

Reviewer #2 (Comments for the Author):

The authors have revised the paper to improve its quality. However, there are still a lot of technical and descriptive errors which may cause misunderstanding and confusion in the manuscript. For example, the authors have emphasized in the Response to Reviewer Comment that shotgun metagenome sequencing were performed only in 13 fecal samples (5 NAFL and 8 NASH patients, biopsy proved), however, in page 19, line 364, the authors still stated "We analyzed the metagenomic sequencing data from "79 NAFLD patients", and the results revealed that glucose metabolism, fatty acid metabolism pathways..." More importantly, it is pointless to conduct correlation study between the microbial functional genes and microbial taxa from the shotgun metagenome sequencing. These correlations has nothing to do with NAFLD biology.

Staff Comments:

Preparing Revision Guidelines

Please return the manuscript within 60 days; if you cannot complete the modification within this time period, please contact me. If you do not wish to modify the manuscript and prefer to submit it to another journal, please notify me of your decision immediately so that the manuscript may be formally withdrawn from consideration by Microbiology Spectrum.

Response to Reviewer 1 Comments

Point 1: The authors carefully addressed my previous questions and concerns.

Response 1: We appreciate the reviewer's comments.

Response to Reviewer 2 Comments

Point 1: The authors have emphasized in the Response to Reviewer Comment that shotgun metagenome sequencing were performed only in 13 fecal samples (5 NAFL and 8 NASH patients, biopsy proved), however, in page 19, line 364, the authors still stated "We analyzed the metagenomic sequencing data from "79 NAFLD patients", and the results revealed that glucose metabolism, fatty acid metabolism pathways..." More importantly, it is pointless to conduct correlation study between the microbial functional genes and microbial taxa from the shotgun metagenome sequencing. These correlations has nothing to do with NAFLD biology.

Response 1: We are thankful for the reviewer's valuable questions and suggestions.

In this study, we examined a total of 129 stool samples from children. These samples included 35 from healthy individuals, 79 from patients with NAFLD, and an additional 15 from follow-up NAFLD patients. Out of these, 35 samples from healthy individuals and 79 from NAFLD patients underwent 16S rRNA sequencing. Furthermore, among the 79 NAFLD samples, including the 13 samples that underwent biopsy investigation, we conducted metagenome sequencing to better understand the correlation between the gut microbial composition and the clinical diagnosis of NAFLD.

We also delved into the disruption of gut microflora associated with specific metabolic pathways in NAFLD. Specifically, we focused on several metabolic pathways relevant to NAFLD, such as glucose metabolism, fatty acid metabolism, short-chain fatty acid (SCFA), and folate pathways. Using metagenomic sequencing data from the 79 NAFLD patients, we conducted an analysis to determine the correlation between bacterial taxa and these metabolic pathways. More details can be found in the section starting at line 364 on page 19. We have made revisions to enhance the clarity of these points.

Once again, we sincerely value your feedback, which has greatly improved our manuscript.

October 9, 2023

Dr. Liming Wang
Institute of Microbiology Chinese Academy of Sciences
No.1 West Beichen Road, Chaoyang District, Beijing, China 100101
Beijing
China

Re: Spectrum01006-23R2 (Role of intestinal flora in the development of children with nonalcoholic fatty liver disease)

Dear Dr. Liming Wang:

Thank you for submitting your manuscript to Microbiology Spectrum. As you will see your paper is very close to acceptance. Please modify the manuscript along the lines the reviewer has recommended. As these revisions are quite minor, I expect that you should be able to turn in the revised paper in less than 30 days, if not sooner. And your manuscript may not be sent for additional external review.

When submitting the revised version of your paper, please provide (1) point-by-point responses to the issues raised by the reviewers as file type "Response to Reviewers," not in your cover letter, and (2) a PDF file that indicates the changes from the original submission (by highlighting or underlining the changes) as file type "Marked Up Manuscript - For Review Only". Please use this link to submit your revised manuscript. Detailed instructions on submitting your revised paper are below.

Link Not Available

Sincerely,

Wei-Hua Chen

Reviewer comments:

Reviewer #1 (Comments for the Author):

The authors carefully addressed my previous questions and concerns.

Reviewer #2 (Comments for the Author):

Since the metabolic pathways identified by shotgun metagenome sequencing of fecal samples primarily come from gut microbiota. These pathways may not be directly linked to NAFLD pathogenesis. Please revise the legend of Figure 6 as Abundance of species significantly correlated with "microbial metabolic pathways potentially related to NAFLD" at the species level.

Preparing Revision Guidelines

Please return the manuscript within 60 days; if you cannot complete the modification within this time period, please contact me. If you do not wish to modify the manuscript and prefer to submit it to another journal, please notify me of your decision immediately so that the manuscript may be formally withdrawn from consideration by Microbiology Spectrum.

Response to Reviewer 1 Comments

Point 1: The authors carefully addressed my previous questions and concerns.

Response 1: We appreciate the reviewer's comments.

Response to Reviewer 2 Comments

Point 1: Since the metabolic pathways identified by shotgun metagenome sequencing of fecal samples primarily come from gut microbiota. These pathways may not be directly linked to NAFLD pathogenesis. Please revise the legend of Figure 6 as Abundance of species significantly correlated with "microbial metabolic pathways potentially related to NAFLD" at the species level.

Response 1:

We appreciate the feedback and have thoroughly considered the point raised.

We have revised the legend of Figure 6 according according to reviewer's comments. The revised legend is "Figure 6. Abundance of species significantly correlated with microbial metabolic pathways potentially related to NAFLD at the species level (n = 79, * $P < 0.05$, ** $P < 0.01$)". This line can be found at line 378 on page 19 in the manuscript.

Thank you for bringing this to our attention, and we hope that this revision adequately addresses the concern.

Re: Spectrum01006-23R3 (Role of intestinal flora in the development of children with nonalcoholic fatty liver disease)

Dear Dr. Liming Wang:

Your manuscript has been accepted, and I am forwarding it to the ASM production staff for publication. Your paper will first be checked to make sure all elements meet the technical requirements. ASM staff will contact you if anything needs to be revised before copyediting and production can begin. Otherwise, you will be notified when your proofs are ready to be viewed.

Sincerely,
Wei-Hua Chen
Editor
Microbiology Spectrum